# Comparative site-specific N-glycoproteome analysis reveals aberrant N-glycosylation and gives insights into mannose-6-phosphate pathway in cancer

Minyong Chen [1✉], Diego M. Assis[2], Matthieu Benet[1], Colleen M. McClung[1], Elizabeth A. Gordon[2], Shourjo Ghose[2], Steven J. Dupard[1], Matthew Willetts[2], Christopher H. Taron[1] & James C. Samuelson [1✉]

N-glycosylation is implicated in cancers and aberrant N-glycosylation is recognized as a hallmark of cancer. Here, we mapped and compared the site-specific N-glycoproteomes of colon cancer HCT116 cells and isogenic non-tumorigenic *DNMT1/3b* double knockout (DKO1) cells using Fbs1-GYR N-glycopeptide enrichment technology and trapped ion mobility spectrometry. Many significant changes in site-specific N-glycosylation were revealed, providing a molecular basis for further elucidation of the role of N-glycosylation in protein function. HCT116 cells display hypersialylation especially in cell surface membrane proteins. Both HCT116 and DKO1 show an abundance of paucimannose and 80% of paucimannose-rich proteins are annotated to reside in exosomes. The most striking N-glycosylation alteration was the degree of mannose-6-phosphate (M6P) modification. N-glycoproteomic analyses revealed that HCT116 displays hyper-M6P modification, which was orthogonally validated by M6P immunodetection. Significant observed differences in N-glycosylation patterns of the major M6P receptor, CI-MPR in HCT116 and DKO1 may contribute to the hyper-M6P phenotype of HCT116 cells. This comparative site-specific N-glycoproteome analysis provides a pool of potential N-glycosylation-related cancer biomarkers, but also gives insights into the M6P pathway in cancer.

[1] New England Biolabs, 240 County Road, Ipswich, MA 01938, USA. [2] Bruker, 40 Manning Road, Billerica, MA 01821, USA. ✉email: chenm@neb.com; samuelson@neb.com

N-glycosylation, one of most abundant protein post-translational modifications, is implicated in the development and progression of many types of cancer. Aberrant protein N-glycosylation often occurs during malignant transformation. Such aberrant N-glycosylation of a protein may include appearance or disappearance of N-glycosylation sites ('N-glycosites'), changes of N-glycan structures at a specific N-glycosite, and changes in up- or down-regulation of a site-specific N-glycosylation. Each of these factors may affect N-glycoprotein activity, stability, and ability to interact with other molecules. Mapping, analysis, and comparison of N-glycoproteomes of cancer cells and their corresponding non-tumorigenic cells can help elucidate cancer mechanisms, identify potential cancer therapeutic targets, and identify glycosylation cancer biomarkers[1].

From an analytical standpoint, N-glycosylation may be evaluated multiple ways. Commonly, N-glycans are structurally assessed following their enzymatic release from glycoproteins (e.g., 'glycomics'). Additionally, N-glycosites may be individually mapped. Finally, N-glycosylation may be addressed using analysis of intact N-glycopeptides (e.g., 'N-glycoproteomics'). The first two approaches are useful, but each alone does not yield both N-glycan structure and glycosite identification. In contrast, bottom-up N-glycoproteomic analysis gives a more data-rich view of N-glycosylation. It not only reveals both N-glycan structure and identifies N-glycosites, but also identifies the repertoire of glycan structures that may occupy each N-glycosite.

Despite many recent improvements in the capabilities of mass spectrometry, analysis of intact N-glycopeptides for N-glycoproteomics is still an analytical challenge for LC-MS/MS. Challenges include variation in electrospray ionization or chromatographic separation of N-glycopeptides, and their structural elucidation using collision-induced dissociation experiments. In this work, we used a novel workflow that combines N-glycopeptide sample enrichment and parallel accumulation serial fragmentation (PASEF) on a trapped ion mobility (TIMS) quadrupole time-of-flight mass spectrometer (timsTOF) to profile intact N-glycopeptides from complex samples (e.g., cell lysates). For N-glycopeptide enrichment, we used an engineered variant of the Fbs1 lectin (Fbs1-GYR), a protein that binds to the conserved N-glycan trimannosyl core (Man3GlcNAc2) of complex and high-mannose N-glycans with submicromolar binding affinity[2]. Prior use of Fbs1-GYR in N-glycopeptide enrichment outperformed established lectin enrichment methods and gave a deeper and greater coverage of the human serum N-glycoproteome[2,3]. The second part of our workflow involved the use of timsTOF to analyze enriched N-glycopeptides. This approach is advantageous because it provides two dimensions of separation for generated ions (HPLC and dual-TIMS). In dual-TIMS analysis, ions from HPLC separation are accumulated and concentrated in the first TIMS section, and then enter second TIMS for additional separation based on ion shape and charges. Ions with same $m/z$ but with different shapes can be separated in the second TIMS section. This is especially useful for separation of N-glycopeptides from non-N-glycopeptides since the shape of a linear peptide is quite different from a 2-D shape of a N-glycopeptide with a N-glycan appended. After this additional separation, the ions of glycopeptides and non-glycopeptides are released for MS/MS fragmentation. Dual-TIMS thus greatly improves MS/MS sensitivity and permits larger scale proteomics.

In this study, we mapped and compared N-glycoproteomes of HCT116 cancer cells and HCT116 cells with a double knockout of two DNA methyltransferases (*DNMT1* and *DNMT3b*) (DKO1)[4]. HCT116 is a human colorectal cancer cell line initiated from an adult male. Upon implantation to immune-compromised mice, the cells form tumors and metastases. Compared to HCT116, DKO1 cells only retain approximately 5% of global genomic DNA methylation[4]. In contrast to HCT116 cancer cells,

DKO1 cells lose both in vitro and in vivo tumorigenicity[5] and are considered isogenic non-tumorigenic versions of HCT116 cells. Thus, to gain insights into glycoproteomic variation between these matched cell lines, the N-glycoproteomes of HCT116 and DKO1 cells were mapped and compared using Fbs1-GYR N-glycopeptide enrichment and timsTOF analysis.

A significant difference in the amount of mannose-6-phosphate (M6P) modification of N-glycans produced by HCT116 and DKO1 cells was observed in this study. M6P N-glycans serve as the signal to sort lysosomal hydrolases to lysosomes[6]. The M6P pathway is involved in tumor formation since the major receptor of M6P, 300 kDa cation-independent M6P receptor (CI-MPR, P11717|MPRI) is downregulated in many tumors and considered as a tumor suppressor[7,8]. The M6P modification is added to the lysosomal hydrolases by two sequential steps. The first step is transferring GlcNAc-1-phosphate from UDP-GlcNAc to high-mannose N-glycans, which is mediated by GlcNAc-phosphotransferase. In the second step, N-acetylglucosamine-1-phosphodiester α-N-acetyl-glucosaminidase removes the GlcNAc portion from GlcNAc-1-phosphate-high mannose N-glycan and thereby forms M6P[9]. In the sorting process, two M6P receptors, CI-MPR and 46 kDa cation-dependent M6P receptor (CD-MPR, P20645|MPRD) recognize and bind to M6P N-glycans on the lysosomal hydrolases and capture these hydrolases into endosomes. The endosomes then fuse to lysosomes and release the hydrolases. Once in the lysosomes, two phosphatases (ACP2 or ACP5) quickly dephosphorylate M6P N-glycans to form high-mannose N-glycans[10]. Thus, in most normal cells, the M6P level is very low. In this sorting process, CI-MPR plays the major role, while CD-MPR helps a smaller subset of proteins to be sorted to lysosomes. Another function of CI-MPR is to recycle some M6P modified lysosomal hydrolases from the extracellular environment back to lysosomes[11]. These proteins escape capture by M6P receptors in the Golgi apparatus and are extracellularly secreted. It has been reported that many cancer cells have increased extracellular secretion of lysosomal hydrolases such as Cathepsin D[12]. However, this mechanism is still unclear. Our N-glycoproteomics and biochemical studies on the M6P pathway provide insights into a potential mechanism.

## Results

**Combination of Fbs1-GYR enrichment and timsTOF enables the identification of large-scale N-glycoproteomes from cell lysates.** The N-glycoproteomes identified in this study reveal site-specific N-glycosylation, which is simultaneous identification of N-glycosites and the appended N-glycans, in HCT116 or DKO1 cells. The workflow of how the N-glycoproteomes were mapped in this study is depicted in Fig. 1a. In the workflow, cell lysates from serum starved HCT116 cells or DKO1 cells were trypsinized to generate total tryptic peptides (pre-enrichment, pre). The N-glycopeptides were enriched by Fbs1-GYR from the total tryptic peptides. The enriched samples (Fbs1) were subjected to timsTOF MS/MS analysis, and the N-glycopeptide spectra were identified by Byonic as described[2,13]. Label-free quantification by counting N-glycopeptide spectrum match (N-glyco PSM) is used to quantify and identify the changes in the N-glycoproteomes between HCT116 and DKO1 cells[14,15].

To evaluate Fbs1-GYR enrichment efficiency from cell lysate samples with low abundance of N-glycoproteins, total pre-enriched tryptic peptide samples were also subjected to timsTOF MS/MS analysis and N-glycopeptide identification by Byonic. Fig. 1b and Supplemental Data 1 and 2 shows Fbs1-GYR enrichment enabled more than one-hundred-fold enrichment of N-glycopeptides. Without Fbs1-GYR enrichment, only 0.28% and

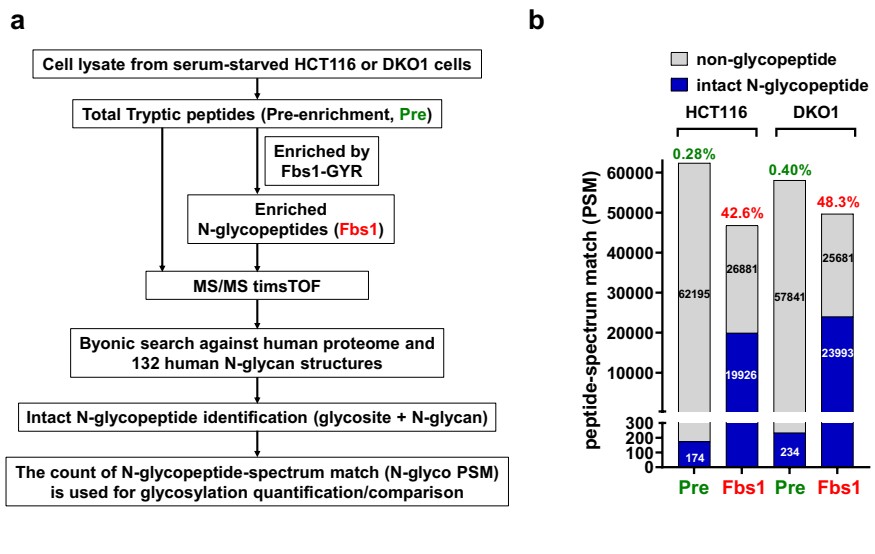

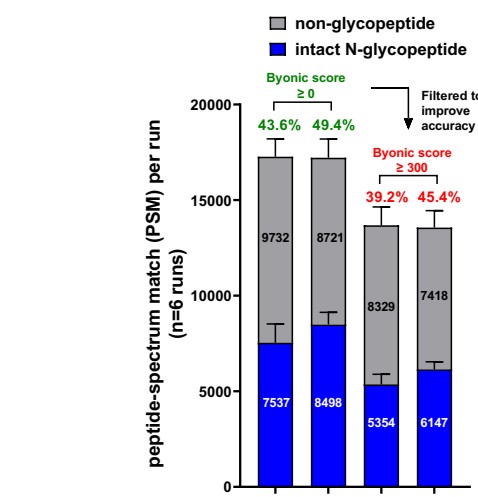

**Fig. 1 Combination of Fbs1-GYR enrichment and timsTOF enables large-scale N-glycoproteome identification from HCT116 and DKO1 cell lysates.**
**a** The workflow for N-glycoproteome identification. **b** Fbs1-GYR enables more than 100-fold N-glycopeptide enrichment. Pre, pre-enrichment, samples were not enriched by Fbs1-GYR; Fbs1, samples were enriched by Fbs1-GYR. The data shown is from MS/MS with 45 min LC. **c** Filtering with Byonic score ≥ 300 improves the MS accuracy. The data shown is a combination of six MS/MS runs with 45 min LC and 90 min LC. Detailed information can be found in Supplemental Data 1–4.

0.40% of total MS/MS spectra were assigned to N-glycopeptide spectra from HCT116 and DKO1 cell lysates, respectively. In contrast, the percentage of N-glycopeptide spectra were improved to 42.6% (for HCT116 cell lysate) and 48.3% (for DKO1 cell lysate), resulting in 152- and 121-fold enrichment, respectively. This data confirms that Fbs1-GYR is an efficient and powerful tool to enrich N-glycopeptides and it is suitable for samples with low abundance of N-glycoproteins such as cell lysates. In this study, a timsTOF MS instrument was used to perform MS/MS analysis. On top of the HPLC separation, which is common in most LC-MS/MS, an additional dimension of ion separation based on ion shapes using the dual trapped ion mobility spectrometry (TIMS) technology enables higher MS/MS resolution[16]. With a combination of Fbs1-GYR enrichment and the timsTOF, a total of 45219 and 50987 N-glycopeptide PSM (N-glyco PSM) were obtained from six MS/MS runs of HCT116 and DKO1 cells, respectively (Fig. 1c and Supplemental Data 3 and 4). In order to more accurately compare the N-glycoproteomes between HCT116 and DKO1, a Byonic score ≥ 300 was used to

remove less confidently assigned N-glycopeptides[17]. After this filtering, a total 32123 and 36880 PSM were assigned as N-glyco PSM from HCT116 and DKO1 samples (six MS/MS runs), respectively (Fig. 1c and Supplemental Data 3 and 4).

**Visualizing N-glycoproteomes of the HCT116 colon cancer cell line and its isogenic non-tumorigenic DKO1 cell line.** Our N-glycoproteomes identified 848 N-glycosites and 2854 unique intact N-glycopeptides from 444 N-glycoproteins in HCT116 cells, and 956 N-glycosites and 3387 intact N-glycopeptides from 507 N-glycoproteins in DKO1 cells (Fig. 2a). Among them, HCT116 and DKO1 share identical 1931 unique intact N-glycopeptides, 665 glycosites and 365 common N-glycoproteins (Fig. 2a).

The detailed N-glycoproteomes of HCT116 and DKO1 are summarized in Supplemental Data 5 (Red tab, "Full summary N-glycoproteomes"). In these N-glycoproteomes, N-glycosylation is organized with an Excel Pivot table. For each N-glycoprotein,

**a**

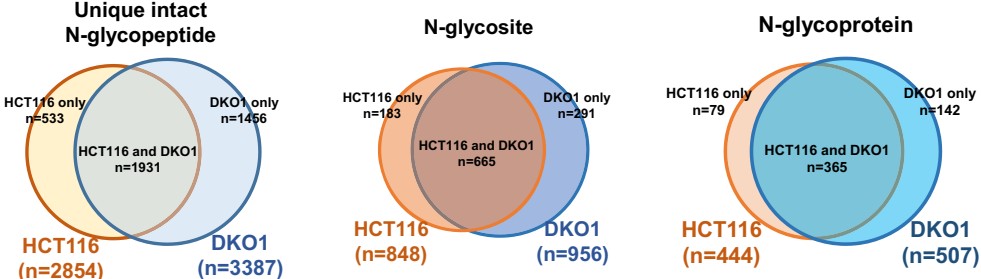

**b**

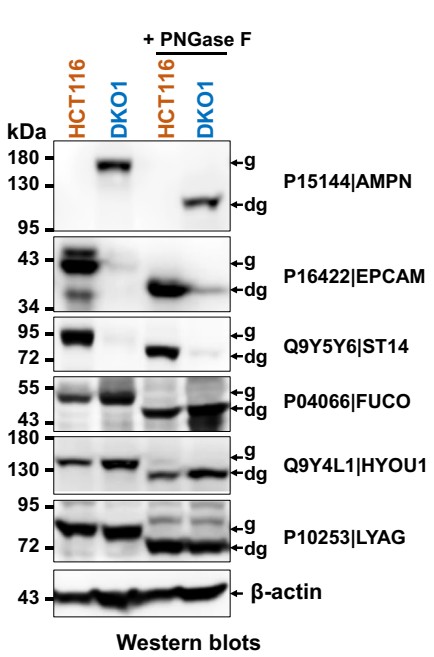

**Western blots**

**c**

| Protein ID | Western blots | | | N-glyco PSM | | |
|---|---|---|---|---|---|---|
| | HCT116 | DKO1 | ratio: H/D | HCT116 | DKO1 | ratio: H/D |
| P15144\|AMPN | 1.00 | 57.54 | H<<D | 0 | 221 | H<<D |
| P16422\|EPCAM | 9.33 | 1.00 | H>>D | 34 | 0 | H>>D |
| Q9Y5Y6\|ST14 | 8.26 | 1.00 | H>>D | 45 | 0 | H>>D |
| P04066\|FUCO | 1.00 | 1.28 | 0.78 | 76 | 97 | 0.78 |
| Q9Y4L1\|HYOU1 | 1.00 | 1.78 | 0.56 | 1269 | 2659 | 0.48 |
| P10253\|LYAG | 1.21 | 1.00 | 1.21 | 855 | 821 | 1.04 |
| P11717\|MPRI (CI-MPR) | 1.00 | 2.53 | 0.40 | 143 | 231 | 0.62 |
| P20645\|MPRD (CD-MPR) | 1.66 | 1.00 | 1.66 | 228 | 134 | 1.70 |

**d**

| Uniprot # \| protein name  N-glycosite  N-glycan | DKO1 | HCT116 | DKO1 | HCT116 |
|---|---|---|---|---|
| | N-glyco PSM | | Relative N-glycosylation abundance | |
| P04066\|FUCO | 97 | 76 | 100.0% | 100.0% |
| N268 | 61 | 51 | 62.9% | 67.1% |
| HexNAc(2)Hex(2) | 1 | | 1.0% | 0.0% |
| HexNAc(2)Hex(3) | 47 | 50 | 48.5% | 65.8% |
| HexNAc(2)Hex(4) | 3 | 1 | 3.1% | 1.3% |
| HexNAc(2)Hex(5) | 5 | | 5.2% | 0.0% |
| HexNAc(2)Hex(6) | 5 | | 5.2% | 0.0% |
| N382 | 36 | 25 | 37.1% | 32.9% |
| HexNAc(2)Hex(4) | 2 | | 2.1% | 0.0% |
| HexNAc(2)Hex(5) | 8 | 5 | 8.2% | 6.6% |
| HexNAc(2)Hex(6) | 25 | 7 | 25.8% | 9.2% |
| HexNAc(2)Hex(6)Phospho(1) | | 13 | 0.0% | 17.1% |
| HexNAc(2)Hex(7) | 1 | | 1.0% | 0.0% |

**e**

| Uniprot # \| prot name  N-glycosite  N-glycan | DKO1 | HCT116 | DKO1 | HCT116 |
|---|---|---|---|---|
| | N-glyco PSM | | Relative N-glycosylation abundance | |
| P04066\|FUCO | 97 | 76 | 100.0% | 100.0% |
| N268 | 61 | 51 | 62.9% | 67.1% |
| HM | 10 | 0 | 10.3% | 0.0% |
| PauciM | 51 | 51 | 52.6% | 67.1% |
| N382 | 36 | 25 | 37.1% | 32.9% |
| HM | 34 | 12 | 35.1% | 15.8% |
| M6P | 0 | 13 | 0.0% | 17.1% |
| PauciM | 2 | 0 | 2.1% | 0.0% |

protein identity (Uniprot # and protein name) is listed on the top and N-glycosites (amino acid position of the N-glycan attaching asparagine (N)) are listed below the protein identity. N-glycan composition pertaining to each glycosite is listed below the glycosite. Next to the protein identity, glycosite, and N-glycan composition, the corresponding N-glyco PSM identified from HCT116 and DKO1 cells are listed.

In order to facilitate the comparison of N-glycoproteomes between HCT116 and DKO1 cells and to more directly visualize the N-glycosylation landscapes, all the 132 N-glycans used in Byonic search were categorized into eight classes based on similar structures and functions: HM (high mannose), Sia (sialylated), C or Hyb (complex or hybrid N-glycans with no sialic acid or fucose modification), Fuc (fucosylated), Sia/Fuc (sialylated and

**Fig. 2 Summary and visualization of the large-scale N-glycoproteomes of HCT116 and DKO1 cells. a** Comparison of overall numbers of unique intact N-glycopeptides, N-glycosites and N-glycoproteins identified from HCT116 and DKO1 cells. **b** Western blots of P15144|AMPN, P16422|EPCAM, Q9Y5Y6|ST14, P04066|FUCO, Q9Y4L1|HYOU1, and P10253|LYAG. "g" and "dg" denote "glycosylated protein" and "deglycosylated protein", respectively. β-actin blot serves as a protein loading control. Representative blots are shown from $n = 3$ experiments. **c** Comparison of the protein relative levels between HCT116 (H) and DKO1 (D) cells determined by Western blots and N-glyco PSM. The Western blots of P11717|MPRI (CI-MPR) and P20645|MPRD (CD-MPR) are shown in Fig. 7c, d. Relative protein levels determined by Western blots were an average from three biological samples. N-glyco PSM of each protein were retrieved from the N-glycoproteomes (Supplemental Data 5). **d**, **e** P04066|FUCO is used as an example to demonstrate differences in site-specific N-glycosylation between HCT116 and DKO1 cells. **d** is a full summary with all N-glycan compositions, and **e** is a simplified summary with N-glycan classes. The green row indicates N-glycoprotein identity (Uniport # and protein name). N-glycosites in the protein are listed below N-glycoprotein identity in yellow rows. Under the N-glycosite, N-glycan compositions are listed (**d**) and (**e**) lists the N-glycan classes attached to each N-glycosite. N-glyco PSM from DKO1 or HCT116 are listed next to protein identities, N-glycosites, and N-glycans. Relative glycosylation abundance of a site-specific N-glycosylation is calculated by N-glyco PSM of a site-specific N-glycosylation divided by the total N-glyco PSM of the corresponding protein. HM, PauciM, and M6P indicate high mannose, paucimannose, and mannose-6-phosphate N-glycans, respectively. The entire site-specific N-glycoproteomes of HCT116 and DKO1 can be found in Supplemental Data 5.

fucosylated), PauciM (paucimannose), M6P (mannose-6-phosphate), and Mono or Di (monoglycan or diglycan) (Supplemental Data 5, Purple tab: "classified N-glycans" and Yellow tab: "glycoproteomes with glycan class")[18,19]. Thus, the N-glycoproteomes can be visualized as a simplified version (Supplemental Data 5, Blue tab: "Simplified N-glycoproteomes"). The simplified version enables more efficient comparison of site-specific N-glycosylation between HCT116 and DKO1.

Furthermore, protein expression level affects the absolute number of N-glyco PSM relating to site-specific N-glycosylation. Thus, to improve the accuracy and fairness of comparison between different cell lines, the site-specific N-glycosylation should be normalized to the protein expression level. The expression level of eight N-glycoproteins in HCT116 and DKO1 cells were detected and quantified by Western blots (Fig. 2b for 6 proteins and Fig. 7c, d for the rest of two proteins). N-glycosylation of these eight proteins was confirmed by their molecular weight mobility shift after PNGase F deglycosylation. The ratios of the protein levels in HCT116 and DKO1 cells determined by Western blots were compared with the ratios of total N-glyco PSM of the proteins in HCT116 and DKO1 cells (Fig. 2c). All eight protein expression levels are positively correlated to their total N-glyco PSM (Fig. 2c). This suggests that in general the total N-glyco PSM of a protein reflects the relative protein level between the cell lines. To offset the effect on the protein expression level affecting the number of specific N-glycosylation, relative glycosylation abundance (RGA) of a site-specific N-glycosylation is calculated by N-glyco PSM of a site-specific N-glycosylation divided by the total N-glyco PSM of the corresponding protein. Thus, RGA of all the site-specific N-glycosylation from HCT116 and DKO1 cells are calculated and included in the N-glycoproteome data sets (Supplemental Data 5).

These mapped large-scale N-glycoproteomes provide the N-glycosylation landscapes in HCT116 and DKO1 cells, including N-glycosylation macro-heterogeneity (glycosite occupancy in a protein) and micro-heterogeneity (N-glycan diversity at a glycosite). N-glycosylation of Alpha-L-fucosidase 1 (P04066|FUCO) is used as an example (Fig. 2d, e). There are two N-glycosites, N268 and N382, which were detected in both HCT116 and DKO1 cells, although three glycosites (N241, N268 and N382) are predicted in FUCO. N268 is the major N-glycosite in both cell lines. N268 in HCT116 is exclusively modified by paucimannose N-glycan with 67.1% RGA in the protein N-glycosylation, while N268 in DKO1 is occupied with 10.3% high mannose N-glycan and 52.6% paucimannose N-glycan. N-382 shows a more significant change in microheterogeneity. In DKO1 cells, N382 is primarily occupied by high mannose N-glycan with 35.1% RGA. In contrast, more than half of the high mannose N-glycan in this site are phosphorylated to form M6P

N-glycan (HexNAc(2)Hex(6)Phospho(1)) (17.1% M6P vs 15.8% HM in this glycosite) in HCT116 cells.

**Comparison of the site-specific N-glycosylation/microheterogeneity between HCT116 and DKO1 cells.** To more efficiently compare the N-glycosylation between HCT116 and DKO1 cells, all the detected N-glycoproteins were divided into four groups shown in the scatter plot in Fig. 3a and Supplemental Data 6, Brown tab. The first group is the proteins with N-glycosylation (N-glyco PSM > 5) detected in both HCT116 and DKO1 cells (green dots). With a relatively high amount of N-glyco PSM, the site-specific N-glycosylation can be compared in this group. The site-specific N-glycosylation of the proteins with N-glyco PSM ≤ 5 in either one of the cell lines cannot be fairly compared. However, they are still valuable especially for those proteins with large differences in N-glycosylation between the two cell lines. Thus, they are further divided into three groups. The HCT116-specific group is comprised of proteins with N-glycosylation in HCT116 at least five time greater than that in DKO1 cells (red dots); the DKO1-specific group is comprised of proteins with N-glycosylation in DKO1 at least five times greater than that in HCT116 (purple dots); and last group is comprised of proteins with an N-glycosylation ratio between HCT116 and DKO1 less than or equal to five (gray dots).

The significant difference in the total N-glyco PSM of the proteins in HCT116-specific or DKO-specific groups suggests overexpression of the proteins in the corresponding cell line. The protein overexpression pattern of three proteins was confirmed by Western blots: P15144|AMPN, P16422|EPCAM, and Q9Y5Y6|ST14 (Fig. 2b). Figure 3b and Supplemental Data 6 (Green tab) list the identities of 28 proteins from HCT116-specific group (red dots in Fig. 3a) and 46 proteins from DKO1-specific group (purple dots in Fig. 3a). Since these N-glycoproteins are overexpressed either in the cancer cell line or non-cancer cell line, they could serve as cancer drivers and potential cancer biomarkers. Indeed, four proteins (P08581|MET, P21860|ERBB3, P04626|ERBB2, and P25445|TNR6) from HCT116-specific group and one protein (P46531|NOTC1) from DKO1-specific group are oncogene products (cancer driver score = 4). P16422|EPCAM and Q95604|1C17 from HCT116-specific group and Q9Y6N7|ROBO1 from DKO1-specific group are also high pathogenic cancer drivers (cancer driver score = 3). The detailed site-specific N-glycosylation of HCT116-specific group and DKO1-specific group can be retrieved from Supplemental Data 5. Here, the site-specific N-glycosylation of P16422|EPCAM is shown as an interesting example (Fig. 3b). EPCAM, aka CD326, is an oncogenic signaling molecule, a novel therapeutic target and cancer stem cell marker[20,21]. Glycosylation at N198 is instrumental to the stability of EPCAM[22]. EPCAM molecules derived from head and neck carcinomas are hyperglycosylated, while EPCAM from healthy head and neck tissues contains no or little

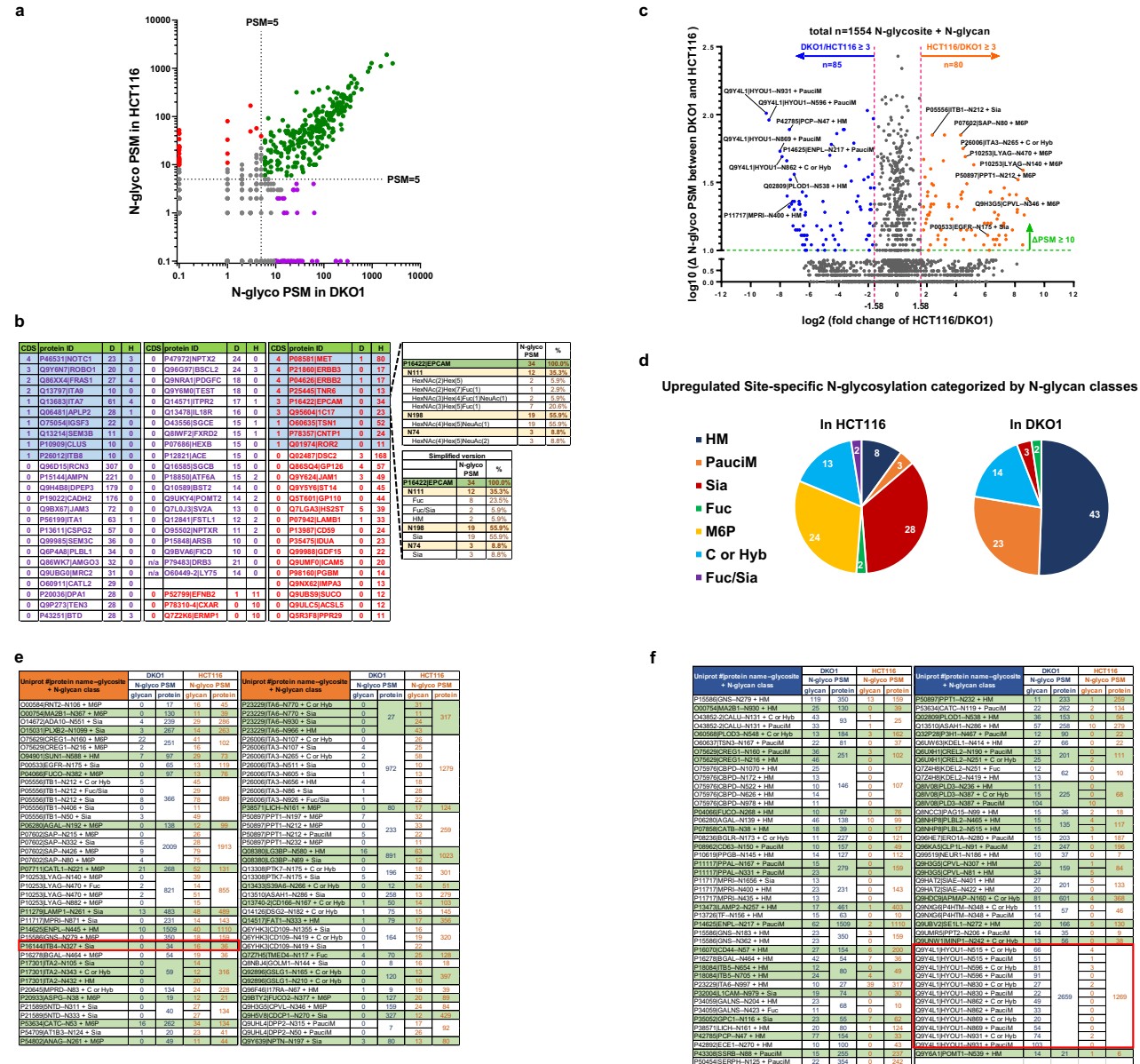

**Fig. 3 Comparative analysis of site-specific N-glycosylation between HCT116 and DKO1 cells. a** Scatter plot to divide all the detected N-glycoproteins into 4 groups based on the N-glyco PSM numbers in HCT116 cells (y-axis) and DKO1 cells (x-axis). The dotted line is a mark of N-glyco PSM = 5. Green dots represent a group of proteins with N-glyco PSM > 5 in both DKO1 and HCT116 cells. The rest of proteins with N-glyco PSM ≤ 5 in one of the cell lines are further divided into three groups. Red dots represent HCT116-specific proteins where N-glycosylation is at least five times greater in HCT116 cells than in DKO1 cells; purple dots represent DKO1-specific proteins where N-glycosylation in DKO1 is at least five times greater than in HCT116 cells; gray dots represent the proteins with N-glycosylation ratio between HCT116 and DKO1 < 5. **b** The N-glyco PSM in HCT116 (H) and DKO1 (D) of the proteins represented by red dots and purple dots (in **a**) are listed. Cancer driver score (CDS) of the corresponding genes is also evaluated. The proteins with CDS ≥ 1 are highlighted in blue. The site-specific N-glycosylation of P16422|EPCAM is presented as an example in these two groups. **c** Volcano plot to show the comparative analysis of site-specific N-glycosylation of the proteins with N-glyco PSM > 5 in both HCT116 and DKO1 cells (green dots in **a**). A total of 1554 site-specific N-glycosylation data points are plotted according to log2 (the fold change of HCT116/DKO1) in x-axis and log10 (the difference of N-glyco PSM between DKO1 and HCT116) in y-axis. The orange and blue dots indicate the upregulated (≥3-fold increase and ≥10 N-glyco PSM difference) site-specific N-glycosylation in HCT116 and DKO1 cells, respectively. The identities of 8 orange or blue dots are indicated as examples. Detailed calculations can be found in Supplemental Data 6, Red tab. **d** Pie charts to categorize by N-glycan classes the upregulated site-specific N-glycosylation in HCT116 and DKO1 cells. The numbers within the pie charts indicate the numbers of upregulated site-specific N-glycosylation in the respective N-glycan class. **e**, **f** list the detailed information of the upregulated site-specific N-glycosylation in HCT116 cells (orange dots in **c**) and DKO1 cells (blue dots in **c**), respectively. Searchable tables can be found in Supplemental Data 6, Orange tab or Blue tab).

N-glycosylation[23]. In this study, N-glycosylation was detected in all three predicted N-glycosites (N74, N111, and N198) in HCT116 cells. N198, the major glycosylation site, is occupied by a sialylated N-glycan (HexNAc(4)Hex(5)NeuAc(1)), and N111 is mainly modified with fucosylated N-glycans. Low protein level

and no N-glycosylation was detected in EPCAM from DKO1 cells. Lack of N-glycosylation may lead to greatly reduced EPCAM protein stability and, thus low protein level in DKO1 cells. High expression level of EPCAM can be one of the drivers that cause HCT116 cells to be tumorigenic.

Next, we compared between DKO1 and HCT116 cells a total of 1554 site-specific N-glycosylations from the proteins with N-glyco PSM > 5 (green dots in Fig. 3a), which is shown in the volcano plot in Fig. 3c (calculation is shown in and Supplemental Data 6, Red tab). In the volcano plot, we use the formula of "Uniprot#|protein name--N-glycosite + N-glycan class" to represent a site-specific N-glycosylation of an individual protein. Using criteria of at least three-fold change of RGA and the difference of N-glyco PSM between DKO1 and HCT116 of at least 10, we scored 80 site-specific N-glycosylations derived from HCT116 cells that are significantly higher than the corresponding N-glycosylation within from DKO1 cells (orange dots, Fig. 3c). Among these 80 changes, the majority are sialylated N-glycan (Sia) (28 changes, 35.0% of total changes), M6P N-glycan (24 changes, 30% of total changes), and complex or hybrid N-glycan (C or Hyb) (13, 16.3% of total changes) (Fig. 3d). We also scored 85 site-specific N-glycosylation derived from DKO1cells that are significantly higher than that from HCT116 (blue dots, Fig. 3c). The majority of the changes in this group are high mannose N-glycan (HM) (43 changes, 50.6% of total changes), paucimannose N-glycans (PauciM) (23 changes, 27.1% of total changes), and complex or hybrid N-glycan (C or Hyb) (14 changes, 16.5% of total changes) (Fig. 3d).

The detailed information of these significant changes is listed in Fig. 3e (or Supplemental Data 6, Orange tab) for those modifications higher in HCT116 cells (orange dots in Fig. 3c) and in Fig. 3f (or Supplemental Data 6, Blue tab) for those modifications higher in DKO1 cells (blue dots in Fig. 3c). Using this comparison, most of the significant site-specific N-glycosylation alterations between HCT116 and DKO1 cells can be easily detected. For example, many N-glycosites in Integrins (N105 of P17301|ITA2; N86, N107, N511 and N605 of P26006|ITA3; N770 and N930 of P23229|ITA6; N50, N212, and N406 of P05556|ITB1; N327 of P166144|ITB4) show elevated sialylated N-glycan modifications in HCT116 cells. Of note, P16144|ITB4--N327 in DKO1 cells does not have any sialic acid modification, while strikingly this site in HCT116 harbors 44.4% (RGA of total ITB4 N-glycosylation) sialic acid modification (Fig. 3e, highlighted in a red box). Another example is Hypoxia upregulated protein 1(Q9Y4L1|HYOU1). All the major N-glycosites in HYOU1, N515, N596, N830, N862, N869, and N931, are modified with significant amounts of paucimannosidic N-glycan (PauciM) and complex or hybrid N-glycans (C or Hyb) in DKO1 cells (Fig. 3f, highlighted in a red box). In contrast, negligible amounts of PauciM and C or Hyb were detected in all these glycosites in HCT116 cells.

The site-specific N-glycosylation shown in Figs. 3e, f provides a great resource to study how site-specific N-glycosylation affects protein function as well as cancer mechanism. It can also be used as a resource for the biomarkers using mass spectrometry (MS)-based detection.

**Comparison of N-glycosylation at the cellular level.** Currently detection of biomarkers using mass spectrometry is not as practical as detection of biomarkers using lectin or antibody-based methods since it requires sophisticated MS instruments and technology[1]. Therefore, we further analyzed and compared the N-glycoproteomes at a level higher than N-glycosite, for example, at the cellular level and protein level.

It is interesting to compare the overall N-glycan content at the cellular level. The categorized N-glycans in HCT116 and DKO1 cells were analyzed as shown in Fig. 4a and Supplemental Data 7 (Red tab). There are no significant differences in overall content of Complex or Hybrid N-glycan (C or Hyb), Fucosylated N-glycan (Fuc) and Mono- or Di-saccharides (Mono or Di)

between HCT116 and DKO1 cells. High mannose N-glycan is slightly more abundant in DKO1 cells (67.62%) than in HCT116 cells (62.16%). Interestingly, HCT116 expresses almost twice more sialylated N-glycans than DKO1 cells (13.62% vs 7.24%). This provides one more piece of evidence to support the notion that hypersialylation occurs in cancer cells[24]. Furthermore, sialylated and fucosylated N-glycans (Sia/Fuc) are approximately 1.4-fold higher in HCT116 than in DKO1 cells. Paucimannosidic N-glycosylation is observed more in DKO1 cells (14.7%) than in HCT116 cells (11.3%). More detailed results on paucimannosidic N-glycosylation are discussed in a following section. The most striking change is mannose-6-phosphate (M6P). M6P in HCT116 and DKO1 was further studied and discussed in a below section.

**Comparison of N-glycosylation at the protein level between HCT116 and DKO1 cells.** Next, we compared N-glycosylation at the protein level between HCT116 and DKO1 cells. The proteins with a significant change in overall N-glycosylation between cancer cells and non-cancer cells can be more practical cancer biomarkers. Fig. 4b and Supplemental Data 7 (Yellow tab) shows the proteins with overexpression of high mannose N-glycans (HM), complex or hybrid N-glycans (C or Hyb), Sialylated N-glycans (Sia), Fucosylated N-glycans (Fuc), or Fucosylated and Sialylated N-glycans (Fuc/Sia) (glycan RGA ratio between HCT116 and DKO1 ≥ 3).

Interestingly, the HM and Sia overexpression patterns are opposite. Most (12 out of 13) of proteins with HM overexpression have higher overall HM content in DKO1 cells, while in contrast, most (11 out 12) of proteins with Sia overexpression have higher overall Sia content in HCT116. In the 13 HM overexpression proteins, 9 of them (P13726|TF, P07858|CATB, Q8NHP8|PLBL2, Q9HAT2|SIAE, P38571|LICH, Q8NCC3|PAG15, O00754|MA2B1, Q9Y2E5|MA2B2, P16278|BGAL) are hydrolases. All the 12 Sia overexpression proteins are annotated to plasma membrane, and 9 of them (P21589|5NTD, P23229|ITA6, P00533|EGFR, P16144|ITB4, P78536|ADA17, P32004|L1CAM, O14672|ADA10, Q9Y639|NPTN, and P18084|ITB5) are involved in cell adhesion. It is known that many cell surface proteins in cancer cells are hypersialylated, which promotes cancer cell metastasis[25]. Here we identified 11 membrane proteins (Q6YHK3|CD109, P54709|AT1B3, P30450|1A26, P21589|5NTD, P23229|ITA6, P00533|EGFR, P16144|ITB4, P78536|ADA17, O14672|ADA10, Q9Y639|NPTN, and P18084|ITB5) including 3 integrins (P23229|ITA6, P16144|ITB4, and P18084|ITB5) which have hypersialylated N-glycans in HCT116 cancer cells. The proteins with C or Hyb overexpression are equally distributed in HCT116 and DKO1 cells, and they are not annotated to a major cluster. Two integrins (P26006|ITA3 and P05556|ITB1) have higher Fuc/Sia modification when they are in HCT116 cells than in DKO1 cells. For Fuc only modification, Q7Z7H5|TMED4 in HCT116 has higher Fuc modification than that in DKO1 cells, while P53801|PTTG shows the opposite pattern.

**Paucimannosidic N-glycosylation in HCT116 and DKO1 cells.** Paucimannosidic N-glycans/paucimannose [$Man_{1-4}Fuc_{0-1}GlcNAc_2$] was initially identified as non-mammalian. Although it is rare in human cells, paucimannose was discovered in neutrophils and some human cancer tissue or cell lines[26–28]. Paucimannosidic N-glycosylation is also gaining more interest since it may be an enriched N-glycosylation signature of human cancers[27].

In the N-glycoproteomes mapped in this study, 11.3% of N-glycosylation in HCT116 cells and 14.7% of N-glycosylation in DKO1 cells were identified as paucimannosidic N-glycosylation (Fig. 4a). Surprisingly, DKO1, the non-tumorigenic cell line, has more paucimannose than its isogenic HCT116 cancer cell line

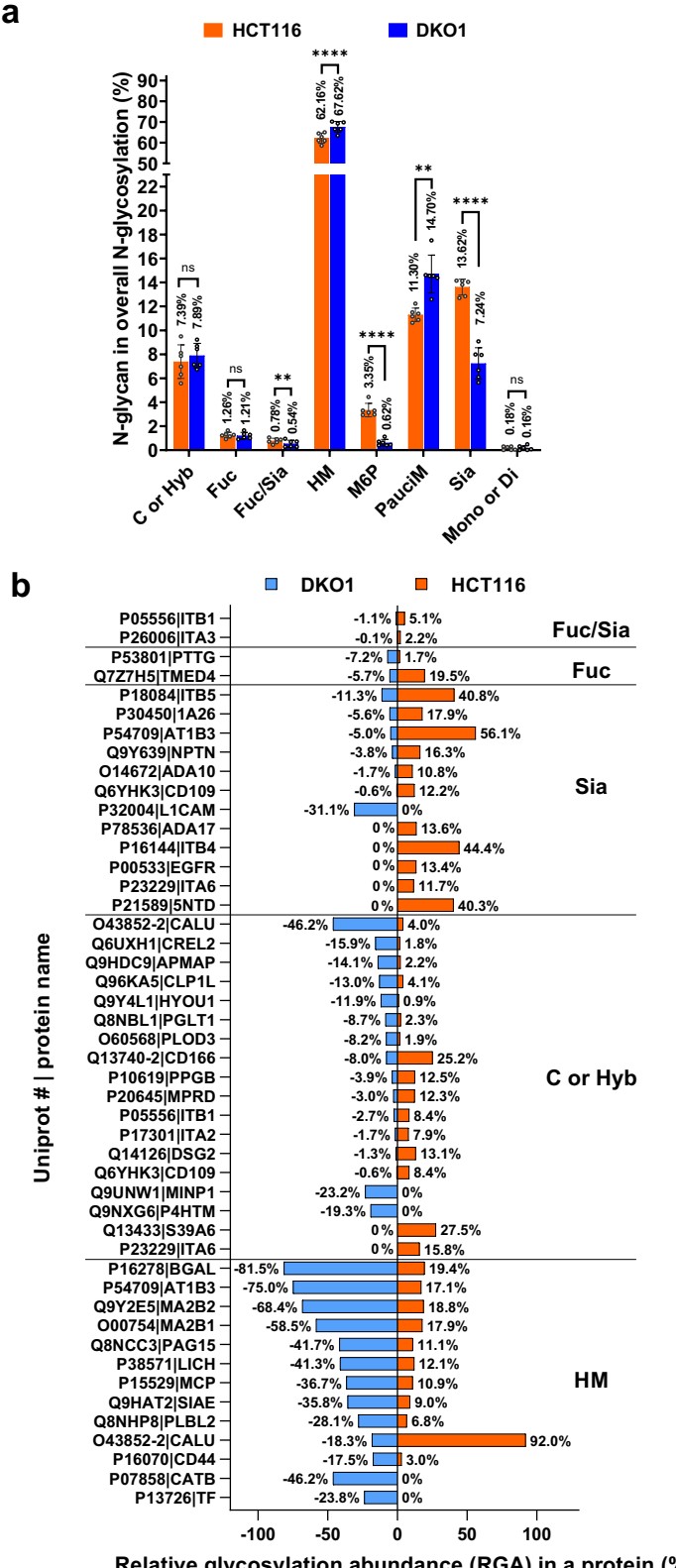

**Fig. 4 Comparative analysis of N-glycosylation at the cell level and the protein level. a** Distribution of 8 N-glycan classes in HCT116 and DKO1 cells. Data were summarized from 6 MS/MS runs. The PSM of a specific N-glycan class in an MS/MS run was normalized to the total N-glyco PSM of the MS/MS run, and the values in percentage were plotted to the *y*-axis. ns: not significant; **, $P \leq 0.01$; ****, $P \leq 0.0001$, mean with SD, t-test, two-tailed, $n = 6$. **b** The N-glycoproteins with significant differences in high mannose N-glycans (HM), complex or hybrid N-glycans (C or Hyb), Sialylated N-glycans (Sia), Fucosylated N-glycans (Fuc), or Fucosylated and Sialylated N-glycans (Fuc/Sia) in HCT116 and DKO1 cells. The glycan ratio between HCT116 and DKO1 is ≥3. The relative glycosylation abundance (RGA) of N-glycans in DKO1 is presented with a negative symbol as a result of the plotting method. Ignore the negative sign and read the absolute value. A more detailed analysis can be found in Supplemental Data 7.

(14.7% vs 11.3%), which suggests that paucimannose described as an N-glycosylation signature of human cancers[27] may not be applied to all the cancer types. Moreover, the N-glycoproteomes allow us to identify the individual N-glycoprotein with high paucimannose modification. Using the criteria of the paucimannose PSM ≥ 10 and RGA ≥ 10% of total N-glycosylation in a protein in either HCT116 or DKO1 cell line, 56 proteins were identified as paucimannose-rich proteins (Fig. 5a and Supplemental Data 8, Green tab). Among them, several proteins are especially of interest including P38571|LICH, O00754|MA2B1, O14773|TPP1, P04066|FUCO, P07602|SAP, P07711|CATL1, P10253|LYAG, P50897|PPT1, Q8IV08|PLD3, Q8NHP8|PLBL2, Q9HAT2|SIAE, and Q9Y6X5|ENPP4, since they display very high percentage of paucimannosidic N-glycosylation (≥50% of N-glycosylation, total N-glyco PSM ≥ 50).

The function of paucimannosidic N-glycosylation in mammalian cells is still largely unclear. To explore its function, the gene ontology annotation to the molecular functions, biological processes, and cellular components of the 56 paucimannose-rich N-glycoproteins was performed using GO Term Mapper (Supplemental Data 8, Blue tab: for Fig. 5b). No major clusters were identified in biological processes terms. 39 out of 56 paucimannose-rich proteins (70%) were annotated to have hydrolase activity in the molecular functions GO terms. More interestingly, in cellular components GO terms, 80% (45 out of 56 proteins) and 73% (41 out of 56 proteins) of the paucimannose-rich proteins were annotated to extracellular exosomes and lysosomes, respectively (Fig. 5b and Supplemental Data 8, Blue tab). This suggests paucimannose may function as one of the signals to target proteins to extracellular exosomes or function as another signal in addition to mannose-6-phosphate (M6P) to sort proteins to lysosomes.

Significant paucimannose changes on 13 individual proteins between HCT116 and DKO1 was also observed (Fig. 5c and Supplemental Data 8, Red tab). Among those proteins, there was no paucimannose detected in 7 proteins in HCT116, while a significant amount of paucimannose was detected in these proteins in DKO1 cells. The most interesting paucimannose change is in Hypoxia up-regulated protein 1 (Q9Y4L1| HYOU1)[29]. 355 PSM out of total 2659 N-glyco PSM (13.4%) are identified as paucimannosidic N-glycosylation in the HYOU1 of DKO1 cells. In comparison, surprisingly, only 1 PSM out of 1269 total N-glyco PSM (0.1%) is paucimannose in the HYOU1 of HCT116 cells. Such a large difference in the same protein also provides strong evidence that paucimannose identification is not due to potential artifacts of mass spectrometry.

It has been reported that N-acetyl-β-D-hexosaminidase α and β (P06865|HEXA and P07686|HEXB, respectively) mediate biogenesis of paucimannose N-glycans in human neutrophils[30]. From the N-glycoproteomes in this study (Supplemental Data 5, search for P06865|HEXA or P07686|HEXB), we have found that P06865| HEXA level in DKO1 (57 N-glyco PSM) is two-fold more than that in HCT116 cells (28 N-glyco PSM), although they show similar N-glycosylation pattern. P07686|HEXB shows a larger difference. 15 and zero N-glyco PSM in DKO1 and HCT116 cells were detected, respectively. The N-glycosylation data suggests the enzyme levels of N-acetyl-β-D-hexosaminidases are remarkably higher in DKO1 cells than in HCT116 cells. This may explain why paucimannosidic N-glycosylation is higher in DKO1 cells than in HCT116 cells.

**Mannose-6-phosphate modification is up-regulated in HCT116 cells as compared to DKO1 cells.** The most significant alteration in N-glycoproteomes between HCT116 and DKO1 cells is the mannose-6-phophate (M6P) modification. M6P N-glycan is a

signal to sort lysosomal enzymes to lysosomes either from Endo Reticulum/Golgi Apparatus or from extracellular matrix[6]. Two receptors, ~300 kDa cation-independent M6P receptor CI-MPR (P11717|MPRI) and ~46 kDa cation-dependent M6P receptor, CD-MPR (P20645|MPRD), recognize M6P modification and then transport the enzymes into lysosomes. Although both receptors are involved in the sorting process and have different subgroups of substrates, CI-MPR plays a major role[31,32]. A portion of CI-MPR, not CD-MPR, is also located on the plasma membrane to recycle the secreted M6P-modified proteins back to lysosomes from the extracellular environment[11]. After lysosomal enzymes reach their destination, lysosomal phosphatases such as ACP2 and ACP5 will quickly dephosphorylate M6P[10]. Thus, M6P modification is very low in normal cells[33]. Our N-glycoproteomic study reveals that the overall M6P modifications detected in HCT116 cells was 5.4-fold more than that in DKO1 cells (Fig. 4a or Fig. 6a). This change was not due to the change of one or two proteins with a high amount of M6P modification. Instead, nearly all N-glycoproteins with M6P modifications (26 out of 28 M6P proteins) contained higher M6P content in HCT116 cells than in DKO1 cells (Fig. 6b, and Supplemental Data 9, Yellow tab), which suggests the activities of some components in the M6P pathway are different in HCT116 and DKO1 cells.

In order to confirm the M6P results from N-glycoproteome studies, immunodetection was employed to compare the M6P level in HCT116 and DKO1 cells. Although a single-chain antibody fragment (scFv) against M6P was developed[34], it shows moderate binding affinity to M6P (kd = 28 μM) as compared to CI-MPR with a binding affinity in the nanomolar range[35]. In this study, soluble CI-MPR (sCI-MPR, available from R&D Systems) containing Domain 1 to 9 was used to measure M6P level in the cells. Among the 9 domains of sCI-MPR, Domains 3, 5, and 9 can specifically recognize and bind to M6P. Radioactive $I^{125}$-labeled sCI-MPR was used to detect glycoproteins containing M6P previously[36]. Here an improved non-radioactive method was developed to detect M6P. sCI-MPR was conjugated with HRP to allow chemiluminescence detection by Western blotting. sCI-MPR-HRP specifically recognizes the M6P epitope in the Western blots since the signal can be competed away by free M6P in a parallel Western blot (Fig. 6c). Expectedly, sCI-MPR-HRP also did not recognize the PNGase F-deglycosylated cell lysate proteins, which is served as further validation (Fig. 6c). The Western blots show that M6P modification in HCT116 cells is 5.6-fold higher than that in DKO1 cells (Fig. 6d), which agrees with 5.4-fold increase results obtained from N-glycoproteomes (Fig. 6a or Fig. 4a). Importantly, this data provides an orthogonal means to prove the accuracy of N-glycoproteomics and reliability of label free quantification by spectral count/PSM count.

In order to explore the possible mechanism that causes hyper-M6P modification in HCT116 cells, N-glycosylation of the M6P receptors (CI-MPR and CD-MPR) was investigated. Both N-glycosylation patterns of CI-MPR and CD-MPR in HCT116 cells were significantly different from those in DKO1 cells (Fig. 7a, b).

CI-MPR glycosylation (Fig. 7a): There are 7 major N-glycosites (N435, N581, N747, N871, N1312, N1656, and N2085) and 6 major N-glycosites (N400, N435, N581, N747, N1312 and N1656) were identified in CI-MPR in the HCT116 cells and DKO1 cells, respectively (the common sites are underlined). No glycosylation at N400 and trace amount of glycosylation at N435 (0.7% with paucimannose) was detected in CI-MPR of HCT116 cells. In contrast, a significant amount of glycosylation at N400 (10.1% with high mannose) and N435 (5.7% with high mannose and 1.3% with paucimannose) was identified in the CI-MPR of DKO1 cells. No sialylated glycosylation was detected in CI-MPR N1656 site in HCT116, while 5.7% sialylated glycosylation was identified in this site in DKO1 cells. On the other hand, no glycosylation at N871

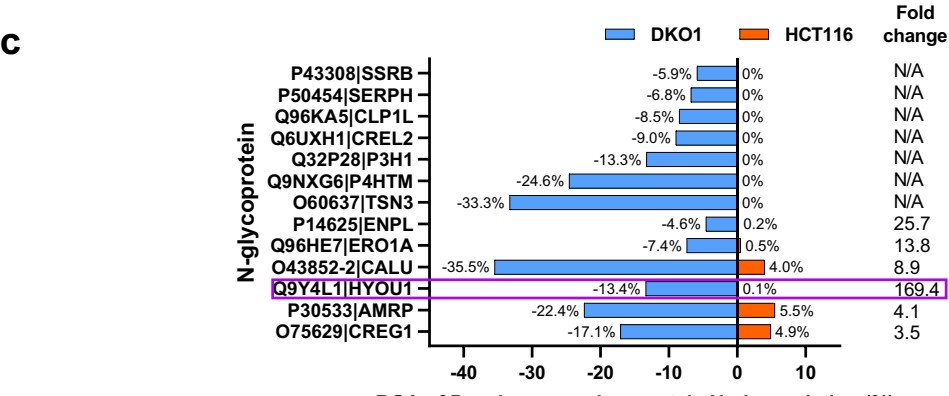

and N2085 was found in CI-MPR of DKO1 cells, while a considerable amount of sialylated N-glycosylation was found at both sites in CI-MPR of HCT116 cells (10.0% at N871 and 2.9% at N2085). Besides, more sialylated glycosylation was observed in CI-MPR N747 site in HCT116 (6.4%) than in DKO1 (2.6%). Overall N-glycan sialylation of CI-MPR is twice more in HCT116 cells (19.3%) than in DKO1 cells (9.3%).

CD-MPR glycosylation (Fig. 7b): There are two N-glycosites (N57 and N83) identified in the smaller M6P receptor, CD-MPR. In DKO1 cells, 72.2 and 27.8% glycosylation were found in CD-MPR at positions N57 and N83, respectively. However, in CD-MPR of HCT116 cells, glycosylation at N57 was decreased to 40%, and glycosylation at N83 was increased to 60%. At the N83 site, more complex or hybrid N-glycans (14.7%) and

**Fig. 5 Paucimannosidic N-glycosylation in HCT116 and DKO1. a** A list of the N-glycoproteins with high paucimannosidic N-glycosylation (≥ 10% of the protein total N-glyco PSM in either HCT116 or DKO1). RGA of paucimannose in DKO1 is presented with a negative symbol as a result of the plotting method. Ignore the negative sign and read the absolute value. **b** The gene ontology analysis of the paucimannose-rich N-glycoproteins with the cellular location indicated. 80% (45 out of 56 proteins tested), and 73% (41 out of 56 proteins tested) of the paucimannose-rich N-glycoproteins are annotated to extracellular exosome (highlighted in a red box) and lysosome (highlighted in a dark green box), respectively. These protein identities are also highlighted in the red box and green box, respectively in (**a**). **c** The N-glycoproteins with a significant difference in paucimannosidic N-glycosylation in HCT116 and DKO1 cells. Q9Y4L1|HYOU1 is highlighted in purple for an example due to its large total N-glyco PSM (2659 in DKO1 and 1269 in HCT116) and a 169.4-fold change in paucimannose glycosylation. A more detailed analysis can be found in Supplemental Data 8.

sialylated N-glycans (41.1%) were identified in CD-MPR of HCT116 cells. Similar to CI-MPR, the overall N-glycan sialylation of CD-MPR is two-fold more in HCT116 cells (44.3%) than in DKO1 cells (20.3%).

Notably, N-glycosylation changes may affect receptor activity or stability in the cells[37,38]. There are significant glycosylation changes in the M6P binding domains of CI-MPR such as Domain 3 (at N400 and N435) and Domain 5 (at N747) as well as the M6P binding domain in CD-MPR (at N83). Whether these glycosylation changes affect the M6P binding activity of CI-MPR or CD-MPR needs further investigation. Here, CI-MPR and CD-MPR stability was evaluated by comparing protein level and mRNA level of the receptors. Shown in Fig. 7c, d, CI-MPR protein level detected by Western blots of HCT116 cells is only 40% of that in DKO1 cells. In contrast, the mRNA transcript level of CI-MPR in both cell lines are similar (relative mRNA level 0.94 in DKO1 vs 1.0 in HCT116 detected by qRT-PCR, Fig. 7e). This suggests that CI-MPR is less stable in HCT116 than in DKO1 cells and the different N-glycosylation profile is most likely the reason for differential CI-MPR stability. On the other hand, both protein level and mRNA transcript level are around 1.6-fold higher in HCT116 cells than those in DKO1 cells, suggesting that stability of CD-MPR is similar in both DKO1 and HCT116 cells (Fig. 7c–e). The higher CD-MPR level may complement the shortage of CI-MPR in HCT116 cells for sorting of some hydrolases to lysosomes.

Thus, a working model is proposed from the receptor glycosylation perspective (Fig. 7f) to elucidate the mechanism of hyper-M6P modification in HCT116 cells. In this model, CI-MPR level or activity or both is significantly reduced in HCT116 cancer cells due to aberrant receptor N-glycosylation (relative to N-glycosylation of CI-MPR in DKO1). Thus, M6P-modified lysosomal enzymes cannot be efficiently transported into lysosomes, where M6P can get dephosphorylated by acidic phosphatases (ACP2/5). This causes M6P-modified lysosomal enzymes to accumulate in the Golgi Apparatus and forces these enzymes to be secreted to the extracellular environment. It has been reported that cancer cells secrete lysosomal hydrolases to the outside of cells to digest extracellular matrix to facilitate metastasis[12]. Besides, cancer cells use the Warburg effect to secrete lactic acid to the extracellular environment, which mimics the acidic conditions optimal for lysosomal hydrolases[39]. Regulation of N-glycosylation of CI-MPR may be one of the mechanisms for cancer cells to secrete lysosomal hydrolases. If this model is correct, then HCT116 cells will secrete more M6P proteins into extracellular matrix than DKO1 cells. To test this, HCT116 and DKO1 cell conditioned media were collected and subjected to M6P Western blots using sCI-MPR-HRP. Fig. 7g, h indicates HCT116 cell conditioned medium contains 2.5-fold more M6P content than DKO1 cell conditioned medium, which suggests that HCT116 cells secrete more M6P proteins into the extracellular environment relative to DKO1 cells. As controls, free M6P competition during Western blotting and PNGase F-deglycosylation of sample proteins abolished the M6P signal. Therefore, the Western blot data supports the proposed model.

## Discussion

In this study, we have successfully mapped and quantitatively compared the site-specific N-glycoproteomes of cancer cells, HCT116 and the isogenic non-cancer cells, DKO1. The combination of Fbs1-GYR enrichment technology and timsTOF enabled us to obtain large scale N-glycoproteomes. With an input of 400 ng Fbs1-GYR enriched peptide samples per MS run, an average of more than 13,000 high quality PSM was obtained from each MS run. Around 40% of the obtained PSM were assigned to N-glycopeptides. This led to large scale N-glycoproteomes by combination of 6 MS runs.

Unlike other enrichment methods such as HILIC in which additional fractionation steps are required[19], Fbs1-GYR offers a simple one-step and efficient enrichment of N-glycopeptides from complex samples with low abundance of N-glycomolecules such as cell lysates. In this study, Fbs1-GYR offered more than 100-fold enrichment and thus enabled large-scale N-glycoproteome mapping. This simple one-step enrichment will improve the throughput of sample processing.

Label-free quantification by spectral counting has been widely used as a common quantitative method in proteomics studies[15]. We adopted this spectral counting (counting N-glyco PSM) to quantify the protein N-glycosylation between HCT116 and DKO1 cells. However, in order to (semi-)quantitatively compare site-specific N-glycosylation between different cell lines, spectral counting is not sufficient alone. Differential expression levels of an N-glycoprotein in different cell lines greatly affects the absolute PSM count of a specific N-glycosylation. Thus, it is important to normalize the PSM number of a specific N-glycosylation to the corresponding protein level to allow fair comparison between cell lines. The protein level was found to be positively correlated to the number of total N-glyco PSM of the protein (Figs. 2b, c, and 7c, d). Thus, we normalized a site-specific N-glycosylation to the total N-glycosylation of the protein (we termed this as relative glycosylation abundance, RGA) in order to offset the effect caused by a differential protein expression level.

N-glycosylation quantification in N-glycoproteomes is difficult to validate. In this study, we were able to orthogonally validate the label-free quantification of M6P modification by M6P Western blots. The relative amount of M6P modifications identified in the N-glycoproteomic data is almost identical to the amount identified by M6P Western blots (Fig. 6).

Our study not only helps to elucidate cancer mechanisms and identification of potential cancer biomarkers, but also benefits the study on how epigenetics regulates N-glycosylation. Several studies[40,41] have provided evidence on protein glycosylation regulated by epigenetics/genomic DNA methylation in cells. Since DKO1 cells lose 95% genomic DNA methylation as compared to the parental cell line HCT116, the comparison of N-glycoproteomes of HCT116 and DKO1 also provides a great resource to further study the relationship between epigenetics and N-glycosylation.

In these mapped N-glycoproteomes, hypersialylation was observed in HCT116 cancer cells as compared to DKO1 non-cancer cells (Fig. 4a, b). Overexpression of sialylated N-glycans in

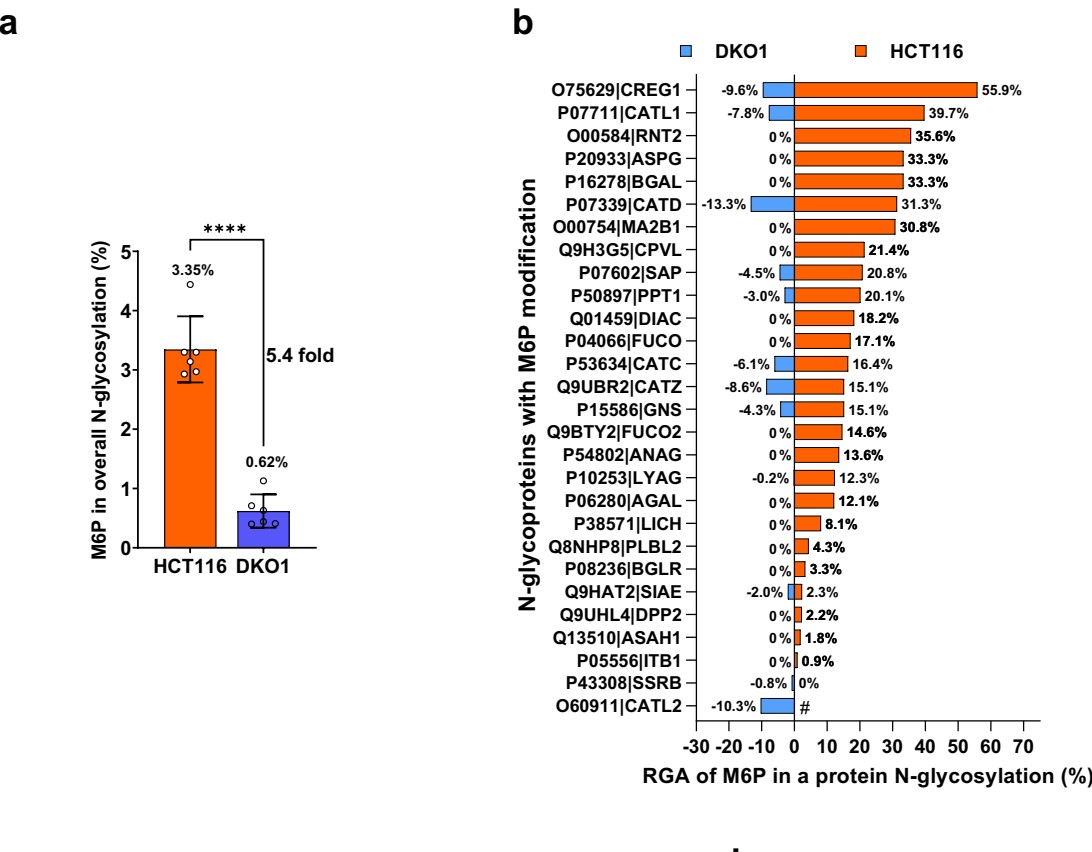

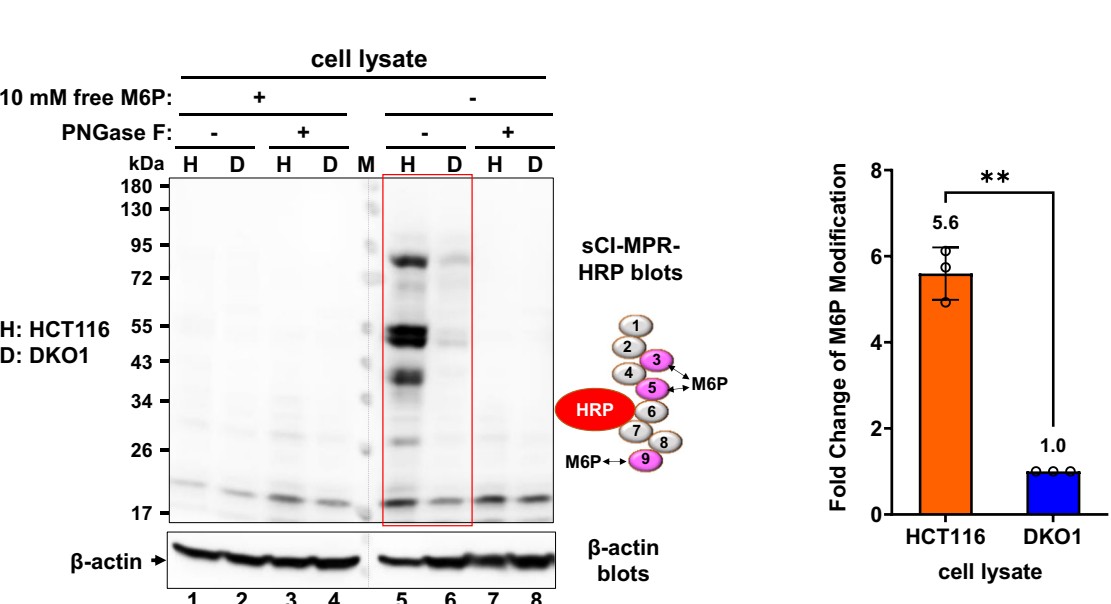

**Fig. 6 M6P modification is more than 5-fold higher in HCT116 cells than that in DKO1 cells.** Comparison of overall (**a**) and individual (**b**) M6P N-glycosylation in HCT116 and DKO1 cells detected by mass spectrometry. Please note (**a**) in this figure is derived from Fig. 4a and is presented to show the correlation between M6P modification levels detected by mass spectrometry and those observed by Western blot analyses shown in (**c**) and (**d**) in this figure. RGA of M6P in DKO1 is presented with a negative symbol as a result of the plotting method. Ignore the negative sign and read the absolute value. #, O60911|CATL2 was not detected in HCT116 cells. **c, d** Western blots show M6P modification is significantly higher in HCT116 cells (H) than in DKO1 cells (D). Representative Western blots are shown in the c), and the quantification summary of M6P modification from three independent Western blots is shown in (**d**). In the Western blot, soluble cation-independent M6P receptor (sCI-MPR) conjugated with HRP (illustrated) was used to probe M6P level in HCT116 and DKO1 cell lysates. The M6P binding domains (Domains 3, 5, and 9) are highlighted in dark red. To demonstrate the specificity of sCI-MPR-HRP, 10 mM free M6P was added to a parallel Western blot to compete off sCI-MPR binding to M6P-modified N-glycoproteins. PNGase F deglycosylated cell lysates were also used as negative controls. β-actin blot serves as a protein loading control. H, HCT116; D, DKO1; M, protein standard marker; **, $P < 0.01$; ****, $P < 0.0001$, mean with SD, t-test, two-tailed, $n = 6$ in (**a**), $n = 3$ in (**d**).

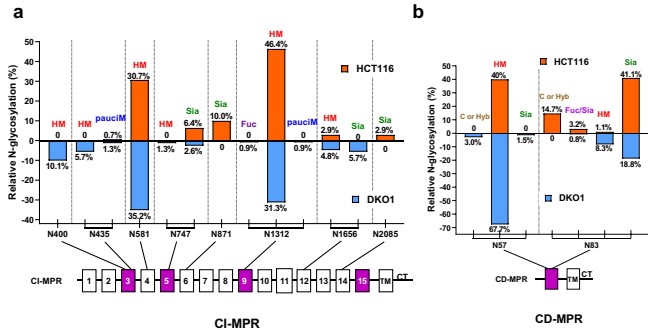

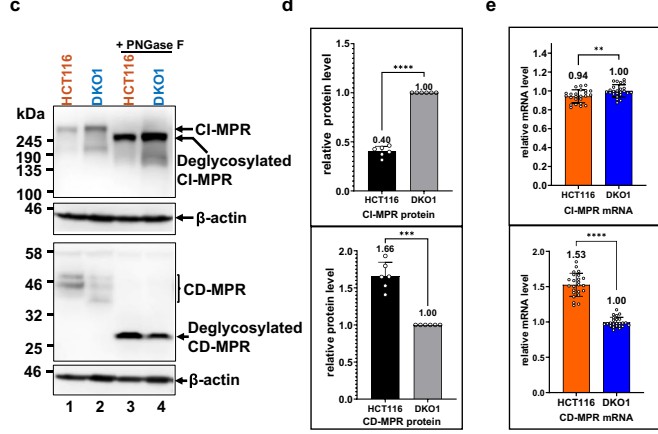

**Fig. 7 Insight into the mechanism of hyper-M6P modification in HCT116 cells relative to DKO1 cells.** Comparison of the site-specific N-glycosylation of M6P receptors, CI-MPR (**a**) and CD-MPR (**b**) in HCT116 and DKO1 cells. 15 extracellular domains, transmembrane domain (TM), and cytosolic tail (CT) are illustrated in CI-MPR. CD-MPR contains one extracellular domain, TM and CT. The domains that interact with M6P moiety are in purple. The N-glycan that modifies the N-glycosite is listed on the top of the bar. **c** Western blots to determine the protein level of CI-MPR and CD-MPR in HCT116 and DKO1 cells. **d** Quantification of the Western blots from three experiments. **e** qRT-PCR to determine the relative mRNA level of CI-MPR and CD-MPR in HCT116 and DKO1 cells. **f** A proposed model to explain hyper-M6P modification in HCT116 cells. In DKO1 cells, the normal function of M6P receptors is to sort M6P-modified lysosomal hydrolases to lysosomes, where M6P gets quickly dephosphorylated by phosphatases (ACP2/5). In contrast, aberrant N-glycosylation of CI-MPR in HCT116 cells may result in lower receptor stability as well as lower affinity to M6P. The lack of CI-MPR function in HCT116 may reroute the M6P-modified lysosomal hydrolases to the secretory pathway, where M6P is not dephosphorylated. Yellow color signifies an acidic environment. **g, h** Western blots show that HCT116 cells secrete more M6P-modified proteins to the medium than DKO1 cells. A representative Western blot is shown in (**g**). A Ponceau S-stained Western blot membrane is to show the protein level in the conditioned media. The quantification summary of M6P modification from three independent Western blots is shown in (**h**). 10× concentrated conditioned medium was probed by sCI-MPR-HRP. Same as in Fig. 6c, Western blots in the presence of 10 mM free M6P and PNGase F deglycosylated conditioned media were used as negative controls. H, HCT116; D, DKO1; M, protein standard marker; *, $P \le 0.05$; **, $P \le 0.01$; ***, $P \le 0.001$: ****, $P \le 0.0001$ mean with SD, t-test, two-tailed, $n = 6$ in (**d**), $n = 24$ in (**e**), $n = 3$ in (**h**).

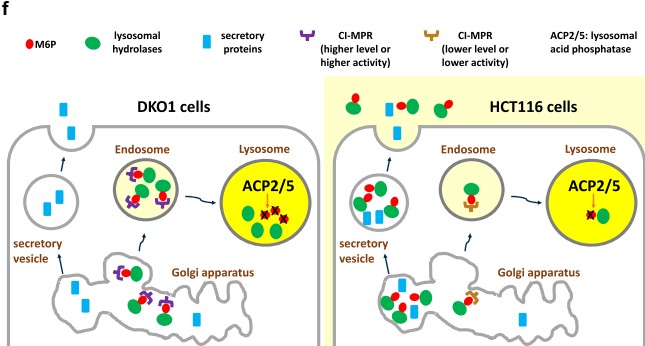

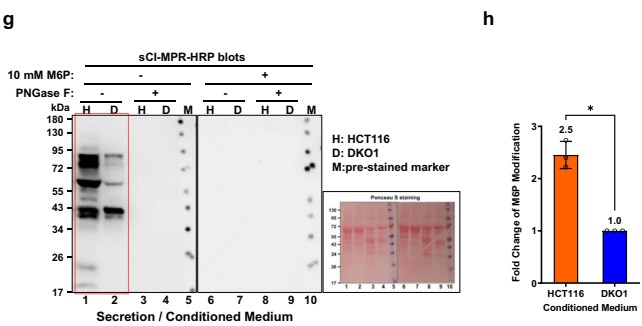

level is around 5-fold lower in HCT116 cells than in DKO1 cells. In addition, the site-specific N-glycosylation of Sialidase-1 in these two cell lines is also different (see Supplemental Data 5, search for Q99519|NEUR1). Sialidase-1 is located in both lysosomes and plasma membrane. The membrane-bound Sialidase-1 can de-sialylate cell surface proteins[24,43]. The reduced level of Sialidase-1 in HCT116 cells may be one of the reasons why hypersialylation happens in HCT116 cancer cells.

Paucimannose modification is quite rare in mammalian cells. However, both HCT116 and DKO1cells display more than 10% paucimannose modification (as a fraction of total N-glycosylation). The function of paucimannose modification in mammalian cells is still not clear. We hypothesize that paucimannose may play a role in guiding proteins to exosomes or lysosomes since 80% and 73% of paucimannose-rich proteins are annotated to exosomes and lysosomes, respectively. Several proteins such as Q9Y4L1|HYOU1 and O75629|CREG1 possess significant higher amount paucimannose modification in DKO1 cells (Fig. 5c). Antibody against paucimannose N-glycans such as Mannitou[44] may be used to detect paucimannose level in such proteins, which may serve as biomarkers to distinguish cancer or non-cancer cells.

M6P N-glycan modification on therapeutic enzymes is one of the key steps in enzyme replacement therapy (ERT)[45]. ERT enzymes modified with M6P N-glycan are recognized and brought to lysosomes by CI-MPR, the major M6P receptor on the plasma membrane. Our profiling on site-specific M6P N-glycan modification on more than 28 enzymes, as well as site-specific N-glycosylation of M6P receptors (CI-MPR and CD-MPR) will help to design better strategies to produce more effective M6P-modified therapeutic enzymes. The Western blot using HRP-conjugated soluble CI-MPR also provides an easy and efficient way to detect M6P modifications on proteins.

cancer cells can be the result of increased level/activity of sialyltransferases, reduced level/activity of sialidases, or a combination of both[42]. In our N-glycoproteomes, no sialyltransferases were observed. However, the N-glycosylation of Sialidase-1 (Q99519|NEUR1) was detected with 37 and 7 total N-glyco PSM in DKO1 and HCT116 cells, respectively. This suggests that Sialidase-1

Several proteins with significant changes in site-specific N-glycosylation are worth more discussion. Additional site-specific N-glycosylation profile information from this study can be retrieved from the Supplemental Data 5 by searching respective Uniprot numbers.

**Calumenins**. O43852|CALU and O43852-2|CALU are isoform 1 and isoform 2 of Calumenin (CALU), respectively. They both have the same length (315 amino acids) and possess only one N-glycosylation site, N131. The only difference in amino acid (aa) sequence of these two isoforms are in the stretch of aa75-aa137, which is $K_{75}$IVSKIDGDK…EYK$N_{131}$ATYGYV$_{137}$ for isoform 1 and $M_{75}$IVDKIDADK…EYR$N_{131}$VTYGTY$_{137}$ for isoform 2. The N131 glycosylation site is within this stretch. However, these two isoforms show very different N-glycosylation behaviors in HCT116 and DKO1 cells. The site-specific N-glycosylation of the isoform 1 in HCT116 and DKO1 is similar to each other. However, the N-glycosylation of isoform 2 in these two cell lines are strikingly different. N131 of isoform2 in DKO1 is modified by 46.24% of Complex or Hybrid N-glycans, 35.48% of Pauci-mannose N-glycans, and only 18.28% of high mannose N-gly-cans, the pattern of which is similar to that of isoform 1 in DKO1 and HCT116. Strikingly, N131 of isoform 2 in HCT116 is mainly occupied by high mannose glycan (92% of total N-glycan). This piece of data suggests the glycosite flanking sequence may affect the N-glycosylation pattern even if the rest of the amino acid sequence is the same.

**5'-nucleotidase**. P21589|5NTD, aka CD73, was found a promis-ing biomarker in cancer patients[46]. Distinct site-specific N-gly-cosylation in DKO1 and HCT116 was found. In DKO1 cells, only one glycosylation site (N311) was detected with major high mannose modification (95% of total N-glycan). In contrast, this protein is hypersialylated in HCT116 cells. Two N-glycosites (N311 and N333) were detected and both were modified with similar amount of sialylated N-glycans (20.1% each of total N-glycan). This discovery may elaborate 5NTD as a more accurate cancer biomarker.

**Epidermal growth factor receptor (P00533|EGFR)**. The EGFR N-glycosylation patterns from HCT116 and DKO1 cells display a significant difference. There are only two N-glycosites, N352 and N361, were detected in DKO1 cells. They are both modified by high mannose glycans (HM), with 61.54 and 38.46% in the total EGFR N-glycosylation in DKO1 cells. In contrast, 4 N-glycosites were detected in HCT116 cells. Similar to EGFR in DKO1 cells, N352 and N361 are occupied by high mannose glycans with abundance of 56.30 and 29.41%, respectively. However, different with EGFR in DKO1 cells, two new glycosites N175 and N603 were detected. Both are mostly modified with a sialylated N-glycan (HexNAc(4)Hex(5)NeuAc(1)) with 10.92% and 2.52% abundance, respectively. Chen et al. have reported that N175 modified with sialylated glycans was detected in lung cancer cells[47]. The N-glycan in this site is important for EFGR activity and it can stabilize the interaction between EGFR and its ligand EGF[48]. The N-glycosylation difference at this site between these paired HCT116 cancer cells and DKO1 non-tumorigenic cells suggests the N-glycosylation at N175 may be used as a cancer biomarker or may contribute to EGFR as an oncogenic driver.

**Hypoxia up-regulated protein 1 (Q9Y4L1|HYOU1)**. This ER chaperone protein (heat shock protein 70 family) is highly expressed in both DKO1 and HCT116 cells with 2659 and 1269 N-glyco PSM, respectively. The N-glycosylation pattern of HYOU1 in HCT116 cells is distinct from that in DKO1 cells. In

HCT116 cells, almost all of N-glycosylation of HYOU1 is high mannose N-glycan (97.8% of total N-glycan), while in DKO1 cells, high mannose content is reduced to 73.4% and other N-glycosylation increases. The increased N-glycosylation is 11.9% with complex or hybrid N-glycan and 13.4% with paucimannose N-glycan. How the change of N-glycosylation affects the activity of HYOU1 needs further investigation.

In summary, combining the Fbs1-GYR N-glycopeptide enrich-ment and Trapped ion mobility spectrometry (timsTOF) enabled mapping of large-scale N-glycoproteomes of HCT116 cancer cells and the isogenic non-cancer DKO1 cells. The N-glycoproteomes were compared from three levels, the N-glycosite level, the protein level, and the cell level. The significant changes in site-specific N-glycosylation provide a molecular basis to elucidate how N-glycosylation affects protein functions. Both HC116 and DKO1 cells contain relatively high percentage paucimannose modification. The identification of paucimannose-rich proteins leads us to hypothesize that paucimannose may guide proteins to exosomes or lysosomes. In HCT116 cells, many proteins were hypersialylated or hyper M6P modified, which may serve as cancer biomarkers. Finally, a model from the M6P receptor N-glycosylation perspective is proposed to explain hyper M6P modification in HCT116 cells. In all, our N-glycoproteomes and systematic comparison provide a great resource for cancer research and for identification of potential therapeutic targets and cancer biomarkers, and especially provides insights into the M6P pathway.

## Methods

**Cell lines and reagents**. HCT116 and its isogenic *DNMT1* and *DNMT3b* double knockout cell line (DKO1) were obtained from Bert Vogelstein's lab[4]. HCT116 was maintained in McCoy5A supplemented with 10% fetal bovine serum and 1% PenStrep, and DKO1 was maintained in the same medium of HCT116 with additional 20 µg/ml Geneticin. Anti-EPCAM (AF960) and Anti-AMPN (AF3815) antibodies were from R&D Systems. Antibodies against β-actin (sc-47778), CD-MPR (sc-365196), ST14 (sc-365482), LYAG (sc-373745), HYOU1 (sc-398224), and FUCO (sc-365496) were from Santa Cruz Biotechnology (SCBT). CI-MPR anti-body (D8Z3J, #15128S) were purchased from Cell Signaling Technology (CST). PNGase F (P0704) and mass spectrometry grade Trypsin (Trypsin-Ultra, P8101) are from New England Biolabs (NEB). Fbs1-GYR proteins were expressed and purified from *E. coli*[2].

**Tryptic peptide preparation from cell lysates**. HCT116 and DKO1 cells were serum-starved for two days in order to reduce N-glycomolecule contamination from bovine serum. Tryptic peptides from the serum-starved cells were prepared using the filter-aided sample preparation (FASP) method as described previously[49] with adjustments for protein release of whole cell samples. Briefly, around 50 µl of pelleted cells (approximately from a 15 cm cell culture plate) were lysed with 800 µl of lysis buffer containing 1% SDS and 100 mM Tris HCl pH 7.5 and were sonicated for 2 min (Qsonica sonicator, 4-tip horn, 30% energy output) to shear the genomic DNA. The cell lysates were clarified by centrifugation at $15,000 \times g$ for 10 min to remove cell debris, and the protein concentration of the cell lysates was then determined using Pierce™ BCA Protein Assay Kit from ThermoFisher (#23225). After 100 mM DTT was added, the cell lysates were then denatured and reduced at 95 °C for 3 min. 400 µl of cell lysates containing 300–400 µg proteins were trans-ferred to a filter unit (Ultracel YM-30, MRCF0R030, MilliporeSigma), and con-centrated to around 100 µl. DTT and SDS were then removed, free thiol groups were alkylated by iodoacetamide (IAA), and tryptic peptides were generated using Trypsin-Ultra (P8101S, NEB) following the protocol described previously[2]. The tryptic peptides were lyophilized and dissolved in LC MS grade water. The con-centrations of tryptic peptides were determined by a Nanodrop spectrophotometer (ThermoFisher) at $A_{280}$ with the sample type option set to 1 Abs = 1 mg/ml.

**N-glycopeptide enrichment by Fbs1-GYR**. The enrichment was performed using the N-glyco-FASP protocol described by Zielinska et al.[49]. 300 µg tryptic peptides (300 µl in 100 mM NH$_4$HCO$_3$) prepared from cell lysate were incubated with 100 µg purified Fbs1-GYR in a Ultracel YM-30 filter unit (30 kDa cutoff, MRCF0R030, MilliporeSigma) at 4 °C for 1 h. The filter units were then centrifuged to remove unbound peptides at $14,000 \times g$ for 10–15 min at 4 °C. To wash away non-specific binding peptides, the peptides/Fbs1-GYR the filter units were then mixed with 300 µl 100 mM NH$_4$HCO$_3$, incubated at 4 °C for 5 min, then centrifuged at $14,000 \times g$ for 10–15 min at 4 °C (discard the flow-through). The washing step was repeated two more times. The peptides retained in the filter units were eluted with $3 \times 200$ µl 50%

formic acid. In the elution step, the peptides/Fbs1-GYR in the filter units were incubated with 200 μl 50% formic acid for 5 min at room temperature, and then was centrifuged at 14,000 × *g* for 10–15 min. The flow-through containing the enriched N-glycopeptides was collected. Repeat this elution step two more times. All the three elution from the same sample was combined and lyophilized overnight to remove formic aid and residual $NH_4HCO_3$. The enriched N-glycopeptides were reconstituted in LC MS grade water with 0.1% Formic Acid, and the concentrations were measured using a Nanodrop Spectrophotometer at $A_{280}$ with the sample type option set to 1 Abs = 1 mg/ml. These Fbs1-GYR enriched peptides were then subjected to timsTOF (Bruker, Billerica, MA) MS/MS analysis.

**timsTOF mass spectrometry analysis**. The pre-enriched tryptic peptides and Fbs1-GYR enriched samples were submitted to MS/MS analysis using a NanoElute system (Bruker) coupled online to a hybrid TIMS-quadrupole TOF mass spectrometer[50] (timsTOF Pro, Bruker) via a nano-electrospray ion source (Captive Spray, Bruker) operating with 3 L/min dry gas at 180 °C and a capillary voltage of 1700 V. Approximately 400 ng of peptides or glycopeptides were separated on an Aurora column 25 cm × 75 μm ID, reversed-phase column (IonOpticks, Australia) at a flow rate of 400 nL/min in an oven compartment heated to 50 °C. The column was equilibrated using 5 volumes of water containing 0.1% formic acid. Two gradients with a linear increase starting from 2 to 37% of acetonitrile containing 0.1% formic acid over either 90 min or 45 min were used in these experiments. For the MS/MS runs with a 45 min LC gradient and a 90 min LC gradient, the mass spectrometer was operated in data-dependent PASEF[51] mode with 1 survey TIMS-MS/6 PASEF MS/MS and 1 survey TIMS-MS/10 PASEF MS/MS scans per acquisition cycle, respectively. Ion mobility range from $1/K0 = 1.6$ to $0.6$ Vs cm$^{-2}$ using equal ion accumulation and ramp time in the dual TIMS analyzer of 100 ms each. MS spectra were acquired within a mass range of $m/z$ 100–3000. Suitable precursor ions for MS/MS analysis were isolated in a window of 2 Th for $m/z < 700$ and 3 Th for $m/z > 700$ by rapidly switching the quadrupole position in sync with the elution of precursors from the TIMS device. We made use of the $m/z$ and ion mobility information to exclude singly charged precursor ions with a polygon filter mask and further used 'dynamic exclusion' to avoid re-sequencing of precursors that reached a 'target value' of 40,000 a.u (45 min LC gradient) or 100,000 a. u (90 min LC gradient). The ion mobility dimension was calibrated linearly using three ions from the Agilent ESI LC/MS tuning mix ($m/z$, 1/K0: 622.0289, 0.9848 Vs cm$^{-2}$; 922.0097, 1.1895 Vs cm$^{-2}$; and 1221.9906, 1.3820 Vs cm$^{-2}$). As a simultaneous acquisition of MS spectra at lower and higher collision energies (collision energy stepping CID) is the ideal approach for selectively yielding fragment ions covering both the glycan and the peptide moieties of glycopeptides, a basic stepping mode with optimized fragmentation and fragment transferring conditions were applied for the tandem MS acquisitions[52]. Thus, each precursor was fragmented with two different energies and collision cell tuning to obtain maximum information. For the 45 min gradient runs, collision energies were set as a linear curve in a mobility dependent manner ranging from 32 eV at $1/K0 = 0.6$ to 64 eV at $1/K0 = 1.6$, with collision cell radiofrequency operating at 1300 V, transfer time of 90 μs and pre-pulse storage of 8 μs (step 1). During step 2, collision energies were set ranging from 40 eV at $1/K0 = 0.6$ to 100 eV at $1/K0 = 1.6$, with collision cell radiofrequency operating at 1600 V, transfer time of 64 μs and pre-pulse storage of 10 μs. For the 90 min gradient runs, collision energies ranged from 20 eV at $1/K0 = 0.6$ to 75 eV at $1/K0$ to 1.6, with collision cell radiofrequency operating at 700 V, transfer time of 25 μs and pre-pulse storage of 8 μs (step 1) and from 35 eV at $1/K0 = 0.6$ to 100 eV at $1/K0$ to 1.6, with collision cell radiofrequency operating at 2000 V, transfer time of 100 μs and pre-pulse storage of 13 μs (step 2). Each MS spectrum was acquired with 50% of time for each defined step and these parameters were applied for all charge states inside the polygon filter mask.

**Byonic search**. N-glycopeptide spectra were searched using Bruker raw data (.d) on Byonic (embedded in PMI-Byos, Protein Metrics Inc., Cupertino, CA) against Uniprot Human Database (UP000005640) and a human N-glycan database containing 132 N-glycan structures (provided by Byonic). Search parameters were set as precursor tolerance to 25 ppm and 0.05 Da for fragments, fully specific trypsin digestion, carbamidomethylation as a fixed modification, N-terminal protein acetylation, methionine oxidation and pyro-Glu as variable modifications and cysteine. The number of peptide-spectrum matches (PSM) were used for glycosylation quantification/comparison. All searches considered only fully tryptic peptides and FDR was adjusted to 1%. PivotTable from Microsoft Excel was used to organize and compare the N-glycoproteomes.

**Sample preparation for Western blots**. Cell pellets were lysed in a denaturing buffer (0.5% SDS and 40 mM DTT) and sonicated for 2 min to reduce viscosity. The serum-free medium conditioned by HCT116 or DKO1 cells (two-day secretion) was concentrated around 10 times in 5 kDa cutoff spin concentrators (Cytiva, Marlborough, MA). Deglycosylated cell lysates or conditioned media were prepared by treatment of PNGase F (New England Biolabs). Samples containing around 30 μg proteins was loaded to SDS-PAGE. The protein gels were blotted to PVDF membrane using Trans-Blot Turbo Transfer System (Bio-Rad, Hercules, CA). The Western blots were imaged by LI-COR C-DiGit Western Blot Scanner and quantified by LI-COR Image Studio Lite software. Prestained protein ladders

(P7712 or P7719, NEB) on the membranes were annotated by WesternSure Pen (LI-COR).

**Conjugation of soluble CI-MPR with HRP and M6P Western blot detection**. Soluble CI-MPR (sCI-MPR) (Recombinant human IGF-II R/IGF2R (aa 43-1365) Protein, CF, Cat #: 6418-GR) was obtained from R&D Systems, Inc. (Minneapolis, MN). HRP Conjugation Kit (Cat #: ab102890) was purchased from Abcam (Cambridge, UK). The conjugation was performed according to Abcam's protocol. 50 μg of lyophilized soluble CI-MPR was reconstituted in 100 μl PBS, and then mixed with 10 μl modifier regent from the HRP conjugation kit. The mixture was then added to a HRP mix vial containing 100 μg HRP. The conjugation was performed at room temperature for 4 h in the dark to produce sCI-MPR-HRP conjugate. After conjugation, 10 μl Quencher reagent was added to quench the non-reacted HRP. Please note no azide should be used to preserve sCI-MPR-HRP conjugate, since azide interferes with HRP chemiluminescent detection. The conjugate required no further purification and was directly used in M6P Western blots. The sCI-MPR-HRP conjugate was 1:500-1:1000 diluted into TBST (150 mM NaCl, 25 mM Tris-HCl pH7.5, 0.1% Tween-20) with 3% BSA and incubated with the 3% BSA-blocked blotted membranes at 4 °C overnight in the absence or presence (as a negative control) of 10 mM free mannose-6-phosphate (M6P) (Sigma). The membranes were then washed three times with TBST (1 × 15 min and 2 × 5 min washes) and were subjected to chemiluminescent detection using luminol as HRP substrate (SuperSignal™ West Pico PLUS Chemiluminescent Substrate, ThermoFisher).

**qRT-PCR to detect mRNA levels of CI-MPR and CD-MPR**. HCT116 and DKO1 cells were seeded in 96-well plates (ten thousand cells per well), serum starved for 2 days, and then lysed using Luna Cell Ready Lysis Module (NEB, #E3032S). The lysed cells were directly subjected to qRT-PCR using Luna Universal One-Step RT-qPCR Kit (NEB, #E3005L). The primer sets used in qRT-PCR to detect CI-MPR and CD-MPR mRNA are 5′-CATTCAGTGGGTGACTCTGTT-3′/ 5′-TGCTCTGGACTCTGTGATTTG-3′ and 5′-GCTCTAGTGAAGAGGCTGAA AC-3′/ 5′-GCACACCCTGAAGATGTAGATG-3′, respectively. β-actin mRNA level (qRT-PCR primers: 5′-GGATGCAGAAGGAGATCACTG-3′/5′-CGATC-CACACGGAGTACTTG-3′) was used as internal control for qRT-PCR. ΔΔCt was calculated to indicate the mRNA level.

**Statistics and reproducibility**. All the statistical analyses were performed using t-test (paired, two-tailed) in Microsoft Excel or Graphpad Prism, and presented as means with SD. In this study, three biological replicates (cells with three different passages) for each cell line (HCT116 or DKO1) were used. For N-glycoproteomic studies, two MS/MS runs were performed for each biological replicate, thus a total of six MS/MS runs for each cell line was analyzed. For Western blots to analyze the protein abundance, both relative quantifications of the glycosylated and de-glycosylated proteins were included in the statistical analysis ($n = 6$ for each cell line). For M6P Western blots, the sample size is 3 (three biological replicates for each cell line) for statistical analysis. For qRT-PCR analysis of mRNA levels of CI-MPR and CD-MPR, the statistical sample size is 24 (8 technical replicates for each biological replicate).

**Bioinformatics**. After processing by Byonic software (embedded in PMI-Byos (Protein Metrics Inc)), MS/MS data were organized by PivotTable of Microsoft Excel. The gene ontology search was performed using Generic GO Term Mapper from Princeton University (https://go.princeton.edu/)[53]. The cancer driver score was evaluated using Oncovar server (https://oncovar.org/)[54].

## Data availability

The uncropped and unedited blot images are available in Supplementary Figs. 1–4. The source data of Figs. 1c, 2c, 4a, 6a, 6d, 7d, e, and h are provided in Supplemental Data 10. The mass spectrometry proteomics data have been deposited to the ProteomeXchange Consortium via the PRIDE[55] partner repository (http://www.ebi.ac.uk/pride) with the dataset identifier PXD035301.

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

## Acknowledgements

We thank Guoping Ren, Pierre Esteve, George Spracklin, Cristian Ruse, Julie Beaulieu, Douglas Oswald, for technical advice and critical reading of the manuscript. We also thank Lawrie Veale, Marshall Bern, and St John Skilton from Protein Metrics for technical support of Byonic search engine. We thank Thomas C. Evans Jr., Richard J. Roberts, Salvatore Russello, James V. Ellard, Donald G. Comb, New England Biolabs, and Bruker for research support.

## Author contributions

M.C. designed research; M.C., D.M.A., M.B., C.M.M., E.A.G., S.G., S.J.D., and M.W. performed research; M.C., D.M.A., M.B., E.A.G., and M.W. analyzed data; and M.C., D.M.A., E.A.G., M.W., C.H.T., and J.C.S. wrote the paper. All authors reviewed the manuscript.

## Competing interests

M.C., C.M.M., C.H.T., and J.C.S. are employees and shareholders of New England Biolabs. D.M.A., E.G., S.G., and M.W. are employees of Bruker. The remaining authors declare no competing interests.
