## [Peer Review File · Communications Biology]

Reviewers' comments:

Reviewer #1 (Remarks to the Author):

Notes:

- Comparison of N-glycoproteomes of HCT116 cells with and without two methyltransferases (DKO1 cells)
 - DKO1 is non-tumorigenic
- Serum starvation to decrease N-glycomolecule contamination
- Started with 300-400 ug protein
- Enriched with Fbs1-GYR
 - Binds to conserved trimannosyl core
 - Previously shown to outperform other lectins
- Normalized to expression level as detected by Western blot
 - Any info about protein abundance by MS?
 - Limited to eight proteins
- Paucimannosidic glycans are less abundant in tumorigenic cell lines (contrary to many other cancers)
 - M6P is up-regulated in tumorigenic cells
 - Confirmed by immunodetection using soluble CI-MPR (receptor)
 - CI-MPR and CD-MPR are more glycosylated in tumorigenic cells, leading to sequestration of M6P-modified enzymes in the Golgi and eventual secretion

Is there any information in the literature about glycan-modifying enzymes in these cell lines and how their trafficking is affected? I think a discussion of that, if available, is warranted. Overall, it's an interesting story from a methods perspective but I'm interested in some of the biological context if it's available.

Reviewer #2 (Remarks to the Author):

Chen et al. reported the "Comparative site-specific N-glycoproteome analysis reveals aberrant N-glycosylation and gives new insights into mannose-6-phosphate pathway in cancer". Overall, the manuscript is well written.

In the introduction part, the authors should explain more about the importance of mannose-6-phosphate pathway in cancer. Meanwhile, the authors should have explained the synthesis pathway of M6P.

Also, in the introduction part, the authors wrote too many conclusions which should be omitted.

Does the 132 N-glycan composition human glycan database include phosphate modification? Also, human N-glycan composition numbers should be much higher than 132 N-glycan composition.

Did the authors use an FDR control in glycan identification? The Byonic software does not seem to support FDR control in glycan identification.

Methods are well written and sufficiently elaborated to reproduce the data.

I can see many tables in figures. These tables should not be placed in the figure, but rather presented as separate tables in the article.

Reviewer #3 (Remarks to the Author):

Review of Communications biology – COMMSBIO-22-2667:

This manuscript performed by Chen M., et al. aims to map and compare the site-specific N-

glycoproteomes of a colon cancer cell, HCT116, and an isogenic non-tumorigenic cell, DKO1. Authors used a carbohydrate binding protein named Fbs-1-GYR to enrich N-glycopeptides and applied them into timsTOF mass spectrometer. A massive MS data was obtained. Authors demonstrated that the N-glycopeptide enrichment by Fbs-1-GYR dramatically increased in total N-glycopeptides more than 100-fold enrichment. Expression levels of some glycoproteins were confirmed by western blotting and the obtained results were consistent to those of label-free quantitative MS. Based on eight classes of N-glycoforms, they showed a significant increase of N-glycopeptides modified mainly in sialylated (Sia) and mannose-6-phosphate (M6P) –forms while a significant decrease in high mannose (HM) and paucimannose (PauciM) –forms in HCT116 cells. The hyper-M6P modification was validated by M6P immunodetection. Moreover, the level of a major M6P receptor, CI-MPR, was down-regulated in HCT116. Finally, they proposed a model of hyper-M6P modification in HCT116 and DKO1 cells to indicate the aberrant receptor N-glycosylation and reduced protein level of CI-MPR would play major roles in M6P-modified lysosomal enzymes. Abnormality may cause these enzymes inefficiently transported into lysosomes but rather to be secreted outside the cells.

These data are very impressive. MS analysis of glycoproteins, especially glycosylation sites and forms, are challenging. Several approaches used in this study including N-glycopeptide enrichment and MS analysis, are reasonable and innovative. Many data were also confirmed/validated. Authors provide tons of original data that are useful for understanding.

However, very minor comments should be concerned.

1. In the method, authors used spectrophotometer at A280 to measure the concentrations of digested peptides. Most of protein work use A280 to measure proteins, not peptides. Why do authors use this wavelength? Why not use other wavelength such as A190 for peptides?
2. In the method and results, is that any particular reason why authors performed six MS/MS runs of each cell? Were they technical replicates obtained from the same sample or the sample from six different passages?
3. In the method and according to Byonic search for MS/MS spectrum, DeltaMod is one of parameter to give a confident modification. Usually, DeltaMod over 10.0 means that there is high likelihood that all modification placements are correct. If authors consider this parameter, please add this information as well.

COMMSBIO-22-2667		
Reviewers' comments		Answers to Reviewers' comments
Reviewer #1 (Remarks to the Author):		
	Notes:	
1	Comparison of N-glycoproteomes of HCT116 cells with and without two methyltransferases (DKO1 cells)	Thank you very much for the summary.
2	DKO1 is non-tumorigenic	
3	Serum starvation to decrease N-glycomolecule contamination	
4	Started with 300-400 ug protein	
5	Enriched with Fbs1-GYR	
6	Binds to conserved trimannosyl core	
7	Previously shown to outperform other lectins	
8	Normalized to expression level as detected by Western blot	
9	Any info about protein abundance by MS?	Although there is MS/MS data of pre-enriched samples (Figure 1b and Supplemental Data 1a), which could be used to estimate N-glycoprotein level/abundance, proteins identified in the pre-enriched samples did not cover all the N-glycoproteins identified in the Fbs1-GYR enriched samples (Figure 1c and Supplemental Data 1.2 and 2). Around 34.5% (451 out of 1307) of N-glycoproteins did not show up in the MS/MS data of the pre-enriched samples. Therefore MS/MS data of pre-enriched samples were not used for protein abundance estimation. Instead, Authors used total N-glyco PSM of a N-glycoprotein to reflect the relative protein abundance between the cancer and non-cancer cell lines. The relative abundance of eight endogenous proteins were quantified by Western Blots and are positively correlated to their total N-glyco PSM (Figure 2b and 2c, and Figure 7b).
10	Limited to eight proteins	Thank you very much for the summary.
11	Paucimannosidic glycans are less abundant in tumorigenic cell lines (contrary to many other cancers)	
12	M6P is up-regulated in tumorigenic cells	
13	Confirmed by immunodetection using soluble CI-MPR (receptor)	
14	CI-MPR and CD-MPR are more glycosylated in tumorigenic cells, leading to sequestration of M6P-modified enzymes in the Golgi and eventual secretion	

15 Is there any information in the literature about glycan-modifying enzymes in these cell lines and how their trafficking is affected? I think a discussion of that, if available, is warranted. Overall, it's an interesting story from a methods perspective but I'm interested in some of the biological context if it's available.	To the Authors' best knowledge, there is no information in the literature which addresses the N-glycan-modifying enzymes in these cell lines. The Authors used the information that was obtained in the identified N-glycoproteomes in this study to explain N-glycosylation changes. In the N-glycoproteomes, the N-glycan biogenesis enzymes such as ALG3, ALG6, OST, FUT8, MGAT 2, ST8SIA4 etc., were not detected. However, Authors have identified several N-glycan modifying enzymes: N-acetyl-β-D-hexosaminidase α and β (P06865 HEXA and P07686 HEXB, respectively), Sialidase-1 (Q99519 NEUR1), as well as CI-MPR (P11717 MPRI) and CD-MPR (P20645 MPRD). Using this information, the Authors were able to explain N-glycosylation changes (please see the following paragraphs). On Page 13 (lines 503-511), the Authors talked about a possible mechanism to explain the difference of paucimannosidic N-glycosylation between the cancer and non-cancer cell lines: "It has been reported that N-acetyl-β-D-hexosaminidase α and β (P06865 HEXA and P07686 HEXB, respectively) mediate biogenesis of paucimannose N-glycans in human neutrophils. From the N-glycoproteomes in this study (Supplemental Data 2, search for P06865 HEXA or P07686 HEXB), we have found that P06865 HEXA level in DKO1 (57 N-glyco PSM) is two-fold more than that in HCT116 cells (28 N-glyco PSM), although they show similar N-glycosylation pattern. P07686 HEXB shows a larger difference. 15 and zero N-glyco PSM in DKO1 and HCT116 cells were detected, respectively. The N-glycosylation data suggests the enzyme levels of N-acetyl-β-D-hexosaminidases are remarkably higher in DKO1 cells than in HCT116 cells. This may explain why paucimannosidic N-glycosylation is higher in DKO1 cells than in HCT116 cells." On Page 16 (lines 636-646), the Authors discussed a possible mechanism to explain hypersialylation in HCT116: "In these mapped N-glycoproteomes, hypersialylation was observed in HCT116 cancer cells as compared to DKO1 non-cancer cells (Figure 4a and 4b). Overexpression of sialylated N-glycans in cancer cells can be the result of increased level/activity of sialyltransferases, reduced level/activity of sialidases, or a combination of both. In our N-glycoproteomes, no sialyltransferases were observed. However, the N-glycosylation of Sialidase-1 (Q99519 NEUR1) was detected with 37 and 7 total N-glyco PSM in DKO1 and HCT116 cells, respectively. This suggests that Sialidase-1 level is around 5-fold lower in HCT116 cells than in DKO1 cells. In addition, the site-specific N-glycosylation of Sialidase-1 in these two cell lines is also different (see Supplemental Data 2, search for Q99519 NEUR1). Sialidase-1 is located in both lysosomes and plasma membrane. The membrane-bound Sialidase-1 can de-sialylate cell surface proteins. The reduced level of Sialidase-1 in HCT116 cells may be one of the reasons why hypersialylation happens in HCT116 cancer cells.
---	---

		On Page 13-15 in the Section "Mannose-6-phosphate modification is up-regulated in HCT116 cells as compared to DKO1 cells", the Authors collected N-glycoproteomic data and biochemical data of CI-MPR and CD-MPR as well as proposed a model to explain hyper-M6P modification in HCT116 cancer cells.
Reviewer #2 (Remarks to the Author):		
1	Chen et al. reported the "Comparative site-specific N-glycoproteome analysis reveals aberrant N-glycosylation and gives new insights into mannose-6-phosphate pathway in cancer". Overall, the manuscript is well written.	Thank you very much for the positive feedback.
2	In the introduction part, the authors should explain more about the importance of mannose-6-phosphate pathway in cancer. Meanwhile, the authors should have explained the synthesis pathway of M6P.	Information about the importance of mannose-6-phosphate pathway in cancer and the synthesis pathway of M6P is added in the last paragraph of Introduction (Page 4, lines 115-121): "M6P pathway is involved in tumor formation since the major receptor of M6P, 300 kDa cation-independent M6P receptor (CI-MPR) is down-regulated in many tumors and considered as a tumor suppressor. The M6P modification is added to the lysosomal hydrolases by two sequential steps. The first step is transferring GlcNAc-1-phosphate from UDP-GlcNAc to high-mannose N-glycans, which is mediated by GlcNAc-phosphotransferase. In the second step, N-acetylglucosamine-1-phosphodiester α -N-acetyl-glucosaminidase removes GlcNAc portion from GlcNAc-1-phosphate-high mannose N-glycan and thereby forms M6P. "
3	Also, in the introduction part, the authors wrote too many conclusions which should be omitted.	Two sentences from the fourth paragraph of Introduction (Page 4, line 112) were deleted: "Generated N-glycoproteomes were then compared from different layers: site-specific layer (microheterogeneity), protein-specific layer as well as cell-specific layer. Many significant N-glycosylation changes detected from the N-glycoproteomes can may be potential glycosylation-based cancer biomarkers and may help to explore cancer mechanisms or therapeutic targets."
4	Does the 132 N-glycan composition human glycan database include phosphate modification?	Among the 132 N-glycan compositions, an N-glycan composition with mannose-6-phosphate (HexNAc(2)Hex(6)Phospho(1)) is included (please refer to Supplemental Data 2, "classified N-glycans" Tab).
5	Also, human N-glycan composition numbers should be much higher than 132 N-glycan composition.	The "Human 132 N-glycan" database was compiled from the Consortium for Functional Glycomics and MALDI-TOF profiles, which represents the common N-glycan composition in human.

6	Did the authors use an FDR control in glycan identification? The Byonic software does not seem to support FDR control in glycan identification.	Byonic doesn't support FDR control in glycan identification. Therefore, authors didn't use an FDR control in glycan identification. Here is the information we obtained from Byonic technical support: 1% FDR control in Byonic means that the entire list of spectrum assignments (peptide, modified peptide, glycopeptide) is estimated to be 1% "false", meaning that it overlaps with the distribution obtained from decoy peptides by 1%. This is the standard target / decoy approach. It is hard to obtain a separate FDR for glycopeptides, because the number of glycopeptide matches can vary from zero to most of the spectra, and there is, to our knowledge, no equivalent of a decoy glycan. Some software tools (e.g., pGlyco) give separate peptide and glycans scores, reflecting the number of peaks matched, but not separate peptide and glycan FDRs.
7	Methods are well written and sufficiently elaborated to reproduce the data.	Thank you very much for the positive feedback.
8	I can see many tables in figures. These tables should not be placed in the figure, but rather presented as separate tables in the article.	The Authors prefer those tables to be placed in the figures to ensure better reading flow. Those tables in the figures have accompanied searchable tables in the Supplemental Datasets.

Reviewer #3 (Remarks to the Author):		
	Review of Communications biology – COMMSBIO-22-2667:	
1	This manuscript performed by Chen M., et al. aims to map and compare the site-specific N-glycoproteomes of a colon cancer cell, HCT116, and an isogenic non-tumorigenic cell, DKO1. Authors used a carbohydrate binding protein named Fbs-1-GYR to enrich N-glycopeptides and applied them into timsTOF mass spectrometer. A massive MS data was obtained. Authors demonstrated that the N-glycopetide enrichment by Fbs-1-GYR dramatically increased in total N-glycopeptides more than 100-fold enrichment. Expression levels of some glycoproteins were confirmed by westernblotting and the obtained results were consistent to those of label-free quantitative MS. Based on eight classes of N-glycoforms, they showed a significant increase of N-glycopeptides modified mainly in sialylated (Sia) and mannose-6-phosphate (M6P) –forms while a significant decrease in high mannose (HM) and paucimannose (PauciM) –forms in HCT116 cells. The hyper-M6P modification was validated by M6P immunodetection. Moreover, the level of a major M6P receptor, CI-MPR, was down-regulated in HCT116. Finally, they proposed a model of hyper-M6P modification in HCT116 and DKO1 cells to indicate the aberrant receptor N-glycosylation and reduced protein level of CI-MPR would play major roles in M6P-modified lysosomal enzymes. Abnormality may cause these enzymes inefficiently transported into lysosomes but rather to be secreted outside the cells.	Thank you very much for the summary.
2	These data are very impressive. MS analysis of glycoproteins, especially glycosylation sites and forms, are challenging. Several approaches used in this study including N-glycopeptide enrichment and MS analysis, are reasonable and innovative. Many data were also confirmed/validated. Authors provide tons of original data that are useful for understanding.	Thank you very much for the positive feedback.

	However, very minor comments should be concerned.	
3	1. In the method, authors used spectrophotometer at A280 to measure the concentrations of digested peptides. Most of protein work use A280 to measure proteins, not peptides. Why do authors use this wavelength? Why not use other wavelength such as A190 for peptides?	Measurement based on the UV absorbance of aromatic amino acids at 280 nm (A_{280}) is a common practice to quantify complex peptide mixtures such as peptide mixtures from cell lysates, since the overall aromatic amino acid contents do not significantly change between different cell lysates. The way to use A_{280} to quantify complex peptide mixtures for LC-MS/MS was also validated in the publication ("Simple Peptide Quantification Approach for MS-Based Proteomics Quality Control", https://doi.org/10.1021/acsomega.0c00080). For a specific single peptide or peptide mixtures with less complexity, A_{280} is not suggested to measure peptide concentrations since the aromatic amino acid content differs from peptide to peptide. In that case, other methods such as measuring absorbance in the deep UV region (190 nm-220 nm) are suggested. However, the measurement at A_{190} - A_{220} is more prone to external influence since many solvents and other chemicals will absorb at this wavelength.
4	2. In the method and results, is that any particular reason why authors performed six MS/MS runs of each cell? Were they technical replicates obtained from the same sample or the sample from six different passages?	The Authors performed six MS/MS runs of each cell line due to the size of the data generated as well as instrument and labor time. These six MS/MS runs for each cell line were from the samples of three biological replicates (cells with three different passages), and two MS/MS runs were performed for each biological replicate. The Authors have added this information in the paragraph of "Statistics and reproducibility" in "Methods and materials" section (on Page 7, lines 257-260): "In this study, three biological replicates (cells with three different passages) for each cell line (HCT116 or DKO1) were used. For N-glycoproteomic studies, two MS/MS runs were performed for each biological replicate, thus a total of six MS/MS runs for each cell line was analyzed."
5	3. In the method and according to Byonic search for MS/MS spectrum, DeltaMod is one of parameter to give a confident modification. Usually, DeltaMod over 10.0 means that there is high likelihood that all modification placements are correct. If authors consider this parameter, please add this information as well.	In this manuscript, the Authors only used Byonic score ≥ 300 as the filter parameter to obtain good peptide-spectrum match (PSM). DeltaMod gives an indication of whether modifications are confidently localized. DeltaMod is important for O-glycoproteomic studies since O-glycan modifications tend to cluster together and it is more difficult to confidently assign an O-glycan modification to a specific Ser/Thr. DeltaMod is less concerned in N-glycoproteomic studies since N-glycosylation consensus motif (N-X-S/T/C) guides the identification of location of N-glycan modification. In datasets of this manuscript, the DeltaMod of 98% PSM with Byonic score ≥ 300 is ≥ 10 . The information regarding Byonic score and DeltaMod can be found in this reference: Byonic: Advanced Peptide and Protein Identification Software, DOI:10.1002/0471250953.bi1320s40. The Authors have included this reference in the revision draft (Reference 22, Page 8, line 298).

REVIEWERS' COMMENTS:

Reviewer #3 (Remarks to the Author):

The revised MS made by Chen M., et al. is impressive. Authors not only addressed my concerns but also added more details in methods and make more clear explanation in their results. I have no further comments.